# TRACK-ON: TRANSFORMER-BASED ONLINE POINT TRACKING WITH MEMORY

**Görkay Aydemir**[1]    **Xiongyi Cai**[3]    **Weidi Xie**[3] †    **Fatma Güney**[1,2] †
[1]Department of Computer Engineering, Koç University    [2]KUIS AI Center
[3]School of Artificial Intelligence, Shanghai Jiao Tong University
gaydemir23@ku.edu.tr    † denotes equal supervision

## ABSTRACT

In this paper, we consider the problem of long-term point tracking, which requires consistent identification of points across multiple frames in a video, despite changes in appearance, lighting, perspective, and occlusions. We target online tracking on a frame-by-frame basis, making it suitable for real-world, streaming scenarios. Specifically, we introduce **Track-On**, a simple transformer-based model designed for online long-term point tracking. Unlike prior methods that depend on full temporal modeling, our model processes video frames causally without access to future frames, leveraging two memory modules —spatial memory and context memory— to capture temporal information and maintain reliable point tracking over long time horizons. At inference time, it employs patch classification and refinement to identify correspondences and track points with high accuracy. Through extensive experiments, we demonstrate that **Track-On** sets a new state-of-the-art for online models and delivers superior or competitive results compared to offline approaches on seven datasets, including the TAP-Vid benchmark. Our method offers a robust and scalable solution for real-time tracking in diverse applications.[1]

## 1    INTRODUCTION

Motion estimation is one of the core challenges in computer vision, with applications spanning video compression (Jasinschi et al., 1998), video stabilization (Battiato et al., 2007; Lee et al., 2009), and augmented reality (Marchand et al., 2015). The objective is to track physical points across video frames accurately. A widely used solution for motion estimation is optical flow, which estimates pixel-level correspondences between adjacent frames. In principle, long-term motion estimation can be achieved by chaining together these frame-by-frame estimations.

Recent advances in optical flow techniques, such as PWC-Net (Sun et al., 2018) and RAFT (Teed & Deng, 2020), have improved accuracy for short-term motion estimation. However, the inherent limitations of chaining flow estimations remain a challenge, namely error accumulation and the difficulty of handling occlusions. To address long-term motion estimation, Sand & Teller (2008) explicitly introduced the concept of pixel tracking, a paradigm shift that focuses on tracking individual points across a video, rather than relying solely on pairwise frame correspondences. This concept, often referred to as "particle video" has been revisited in recent deep learning methods like PIPs (Harley et al., 2022) and TAPIR (Doersch et al., 2023), which leverage dense cost volumes, iterative optimization, and learned appearance updates to track points through time.

Despite the advancements, the existing methods for long-term point tracking face two major limitations. First, they primarily rely on offline processing, where the entire video or a large window of frames is processed at once. This allows models to use both past and future frames to improve predictions but inherently limits their applicability in real-world scenarios (Karaev et al., 2024b; Harley et al., 2022). Second, these approaches struggle with scalability, as they often require full attention computation across all frames, leading to significant memory overhead, especially for long videos or large frame windows. These limitations hinder their use in real-world applications, like robotics or augmented reality, where efficient and online processing of streaming video is crucial.

---

[1]Project page: https://kuis-ai.github.io/track_on

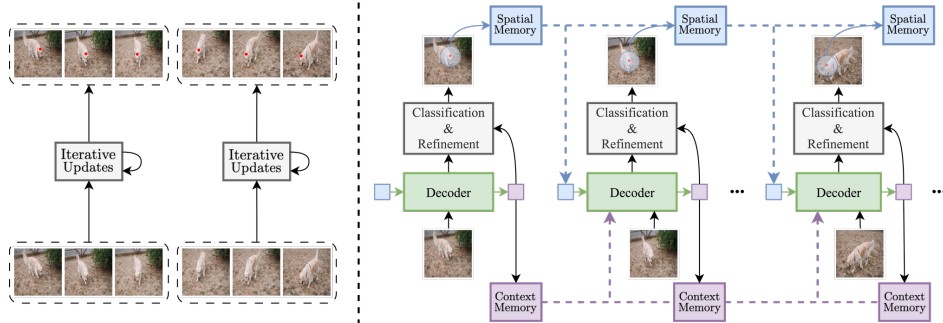

Figure 1: **Offline vs. Online Point Tracking.** We propose an online model, tracking points frame-by-frame (**right**), unlike the dominant offline paradigm where models require access to all frames within a sliding window or the entire video (**left**). In contrast, our approach allows for frame-by-frame tracking in videos of any length. To capture temporal information, we introduce two memory modules: spatial memory, which tracks changes in the target point, and context memory, which stores broader contextual information from previous states of the point.

In this paper, we address the challenge of long-term point tracking in an online processing setting (Fig. 1, right), where the model processes video frames sequentially, without access to future frames (Vecerik et al., 2023). We propose a simple transformer-based model, where points of interest are treated as queries in the transformer decoder, attending the current frame to update their features. Unlike existing methods that aggregate temporal information across video frames, we achieve temporal continuity by updating the query representations with information from two specialized memory modules: spatial memory and context memory. This design enables the model to maintain reliable point tracking over time while avoiding the high computational and memory costs associated with full temporal modeling across entire video sequences.

Specifically, spatial and context memory play distinct but complementary roles. The former aims to reduce tracking drift by updating the query representation with information from the latest frames. This ensures that the query reflects the most recent visual appearance of the tracked point, by storing the content around the model's predictions in previous frames, rather than relying on features of the initial point. On the other hand, context memory provides a broader view of the track's history, storing the point's embeddings from past frames. This allows the model to consider changes to visual content including key information about the point's status, such as whether the point was occluded in previous frames. Overall, spatial memory focuses on positional changes in predictions over time while context memory ensures temporal continuity by providing a full perspective of the track's evolution. Together, these two memory modules aggregate useful temporal information across video.

At training time, the queries from the transformer decoder identify the most likely location by computing embedding similarity with each patch, and are trained using similarity-based classification, akin to contrastive learning. The prediction is then refined by estimating an offset within the local region to find the final correspondence. We conduct extensive experiments demonstrating that our simple patch-classification and refinement approach serves as a strong alternative to the dominant iterative update paradigm (Karaev et al., 2024b; Harley et al., 2022; Doersch et al., 2023). Our method sets a new state-of-the-art among online models and either matches or surpasses offline approaches on seven datasets including the TAP-Vid benchmark.

In summary, our contributions are as follows: (i) a simple architecture that treats points of interest as queries in the transformer decoder, identifying correspondences through patch classification and refinement; (ii) memory modules that effectively store past content and address feature drift in an online manner; (iii) extensive experiments and ablations that demonstrate state-of-the-art performance among online models and competitive results to offline models.

## 2 METHODOLOGY

### 2.1 PROBLEM SCENARIO

Given an RGB video of $T$ frames, $\mathcal{V} = \{\mathbf{I}_1, \mathbf{I}_2, \ldots, \mathbf{I}_T\} \in \mathbb{R}^{T \times H \times W \times 3}$, and a set of $N$ predefined queries, $\mathcal{Q} = \{(t^1, \mathbf{p}^1), (t^2, \mathbf{p}^2), \ldots, (t^N, \mathbf{p}^N)\} \in \mathbb{R}^{N \times 3}$, where each query point is specified

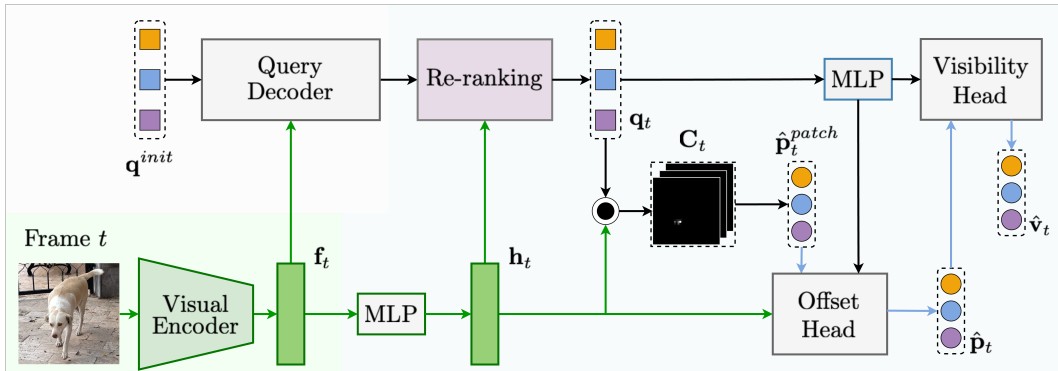

Figure 2: **Overview.** We introduce Track-On, a simple transformer-based method for online, frame-by-frame point tracking. The process involves three steps: (i) **Visual Encoder**, which extracts features from the given frame; (ii) **Query Decoder**, which decodes interest point queries using the frame's features; (iii) **Point Prediction** (highlighted in light blue), where correspondences are estimated in a coarse-to-fine manner, first through patch classification based on similarity, then followed by refinement through offset prediction from a few most likely patches. Note that the squares refer to point queries, while the circles represent predictions, either as point coordinates or visibility.

by the start time and pixel's spatial location, our goal is to predict the correspondences $\hat{\mathbf{p}}_t \in \mathbb{R}^{N \times 2}$ and visibility $\hat{\mathbf{v}}_t \in \{0, 1\}^N$ for all query points in an online manner, *i.e.* using only frames up to the current target frame $t$. To address this problem, we propose a transformer-based point tracking model, that tracks points **frame-by-frame**, with dynamic memories $\mathbf{M}$ to propagate temporal information along the video sequence:

$$\hat{\mathbf{p}}_t, \ \hat{\mathbf{v}}_t, \ \mathbf{M}_t = \Phi\left(\mathbf{I}_t, \ \mathcal{Q}, \ \mathbf{M}_{t-1}; \ \Theta\right) \tag{1}$$

In the following sections, we start by describing the basic transformer architecture for point tracking in Sec. 2.2, then introduce the two memory modules and their update mechanisms in Sec. 2.3.

## 2.2 TRACK-ON: POINT TRACKING WITH A TRANSFORMER

Our model is based on transformer, consisting of three components, as illustrated in Fig. 2: **Visual Encoder** is tasked to extract visual features of the video frame, and initialize the query points; **Query Decoder** enables the queried points to attend the target frame to update their features; and **Point Prediction**, to predict the positions of corresponding queried points in a coarse-to-fine (Doersch et al., 2023) manner.

### 2.2.1 VISUAL ENCODER

We adopt a Vision Transformer (ViT) as visual backbone, specifically, DINOv2 (Oquab et al., 2024), and use ViT-Adapter (Chen et al., 2022b) to obtain dense features at a higher resolution than the standard ViT. We then add learnable spatial positional embeddings $\gamma^s$ to the frame-wise features:

$$\mathbf{f}_t = \Phi_{\text{vis-enc}}\left(\mathbf{I}_t\right) + \gamma^s \in \mathbb{R}^{\frac{H}{S} \times \frac{W}{S} \times D} \tag{2}$$

where $D$ denotes the feature dimension, and $S$ refers to the stride. We use a single-scale feature map from ViT-Adapter for memory efficiency, specifically with a stride of $S = 4$.

**Query Initialization:** To initialize the query features ($\mathbf{q}^{init}$), we apply bilinear sampling to the feature map at the query position $(\mathbf{p}^i)$:

$$\mathbf{q}^{init} = \left\{\text{sample}(\mathbf{f}_{t^i}, \ \mathbf{p}^i)\right\}_{i=1}^{N} \in \mathbb{R}^{N \times D}$$

In practice, we initialize the query based on the features of the start frame $t^i$ for $i$-th query, assuming they can start from different time point, and propagate them to the subsequent frames.

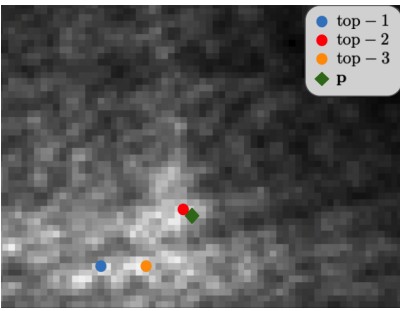

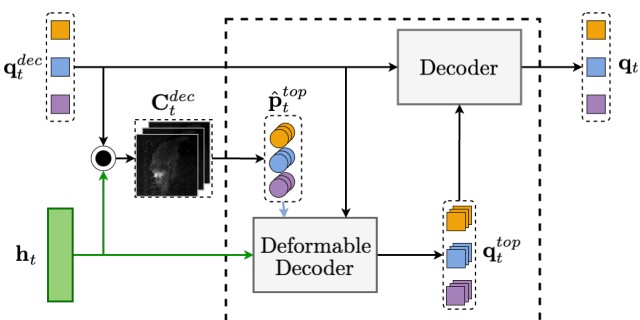

Figure 3: **Top-$k$ Points.** In certain cases, a patch with high similarity, though not the most similar, is closer to the ground-truth patch. The top-3 patch centers, ranked by similarity, are marked with dots, while the ground-truth is represented by a diamond.

Figure 4: **Ranking Module.** The features around the top-$k$ points ($\hat{\mathbf{p}}_t^{top}$) with the highest similarity are decoded using deformable attention to extract the corresponding top-$k$ features ($\mathbf{q}_t^{top}$). These features are then fused with the decoded query $\mathbf{q}_t^{dec}$ using a transformer decoder.

### 2.2.2 QUERY DECODER

After extracting the visual features for the frame and query points, we adopt a variant of transformer decoder (Vaswani et al., 2017), with 3 blocks, *i.e.* cross-attention followed by a self-attention, with an additional feed forward layer between attentions:

$$\mathbf{q}_t^{dec} = \Phi_{\text{q-dec}}\left(\mathbf{q}^{init}, \mathbf{f}_t\right) \in \mathbb{R}^{N \times D} \tag{3}$$

The points of interest are treated as queries, which update their features by iteratively attending to visual features of the current frame with cross attention. These updated queries are then used to search for the best match within the current frame, as explained in the following section.

### 2.2.3 POINT PREDICTION

Unlike previous work that regresses the exact location of the points, we formulate the tracking as a matching problem to one of the patches, that provides a coarse estimate of the correspondence. For exact correspondence with higher precision, we further predict offsets to the patch center. Additionally, we also infer the visibility $\hat{\mathbf{v}}_t \in [0, 1]^N$ and uncertainty $\hat{\mathbf{u}}_t \in [0, 1]^N$ for the points of interest.

**Patch Classification:** We first pass the visual features into 4-layer MLPs, and downsample the resulting features into multiple scales, *i.e.*, $\mathbf{f}_t \to \mathbf{h}_t \in \mathbb{R}^{\frac{H}{S} \times \frac{W}{S} \times D} \to \mathbf{h}_t^l \in \mathbb{R}^{\frac{H}{2^l \cdot S} \times \frac{W}{2^l \cdot S} \times D}$. We compute the cosine similarity between the decoded queries and patch embeddings in four scales, and the similarity map $\mathbf{C}_t^{dec}$ is obtained as the weighted average of multi-scale similarity maps with learned coefficients (details in Appendix Sec. B.2). We then apply a temperature to scale the similarity map and take softmax spatially over the patches within the current frame. The resulting $\mathbf{C}_t^{dec}$ provides a measure of similarity for each query across the patches in the frame.

We train the model with a classification objective, where the ground-truth class is the patch with the point of interest in it. In other words, we perform a $P$-class classification, $P$ is the total number of patches in the frame.

**Re-ranking:** We observed that the true target patch might not always have the highest similarity on $\mathbf{C}_t^{dec}$, however, it is usually among the top-k patches. For example, in Fig. 3, the patch with the second-highest similarity (top-2) is closer to the true correspondence than the most similar patch (top-1). To rectify such cases, we introduce a re-ranking module $\Phi_{\text{re-rank}}$:

$$\mathbf{q}_t = \Phi_{\text{re-rank}}\left(\mathbf{q}_t^{dec}, \mathbf{h}_t, \mathbf{C}_t^{dec}\right) \in \mathbb{R}^{N \times D} \tag{4}$$

where $\mathbf{q}_t$ denotes the refined queries after ranking.

In the re-ranking module (Fig. 4), we identify the top-$k$ patches with the highest similarities and retrieve their corresponding features with a deformable attention decoder. Then, we integrate them

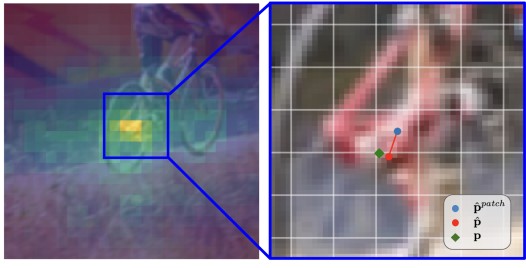

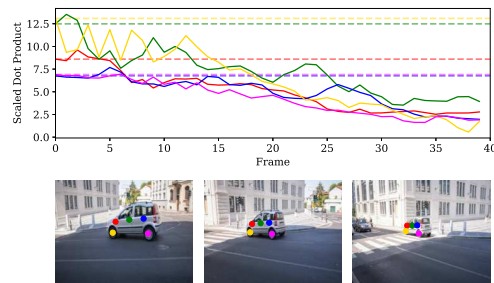

Figure 5: **Offset Head**. Starting with a rough estimation from patch classification (**left**), where lighter colors indicate higher correlation, we refine the prediction using the offset head (**right**). The selected patch center and the final prediction are marked by a blue dot and a red dot, respectively, with the ground-truth represented by a diamond.

Figure 6: **Feature Drift.** For the tracks shown below (start, middle, and final frames), the plot above illustrates the decreasing similarity between the features of the initial query and its correspondences over time, with the initial similarity indicated by horizontal dashed lines.

into the original query features via a transformer decoder to produce refined queries. Using these refined queries, we calculate the final similarity map $\mathbf{C}_t$ and apply a classification loss. Finally, we select the center of the patch with the highest similarity ($\hat{\mathbf{p}}_t^{patch} \in \mathbb{R}^{N \times 2}$) as our coarse prediction. Additionally, we compute an uncertainty score for each top-$k$ location, *i.e.* $\hat{\mathbf{u}}_t^{top} \in \mathbb{R}^{N \times k}$, by processing their corresponding features with a linear layer (see Appendix Sec. B.2 for details).

**Offset Prediction:** For the exact correspondence ($\hat{\mathbf{p}}_t \in \mathbb{R}^{N \times 2}$), we further predict an offset $\hat{\mathbf{o}}_t \in \mathbb{R}^{N \times 2}$ to the patch center by incorporating features from the local region around the inferred patch, as shown in Fig. 5:

$$\hat{\mathbf{o}}_t = \Phi_{\text{off}}(\mathbf{q}_t, \ \mathbf{h}_t, \ \hat{\mathbf{p}}_t^{patch}), \qquad \hat{\mathbf{p}}_t = \hat{\mathbf{p}}_t^{patch} + \hat{\mathbf{o}}_t \qquad (5)$$

Here, $\Phi_{\text{off}}$ is a deformable transformer decoder (Zhu et al., 2021) block with 3 layers, excluding self-attention. In this decoder, the query $\mathbf{q}_t$ is processed using the key-value pairs $\mathbf{h}_t$, with the reference point set to $\hat{\mathbf{p}}_t^{patch}$. To limit the refinement to the local region, the offsets are constrained by the $S$ (stride) and mapped to the range $[-S, S]$ using a tanh activation.

In addition, we predict the visibility $\hat{\mathbf{v}}_t$ and uncertainty $\hat{\mathbf{u}}_t$, using visibility head $\Phi_{\text{vis}}$. We first decode the region around the predicted location $\hat{\mathbf{p}}_t$ (Eq. 5) using a deformable decoder layer. Then, we predict visibility and uncertainty by applying a linear layer to the decoded queries. At training time, we define a prediction to be uncertain if the prediction error exceeds a threshold ($\delta_u = 8$ pixels) or if the point is occluded. During inference, we classify a point as visible if its probability exceeds a threshold $\delta_v$. Although we do not directly utilize uncertainty in our predictions during inference, we found predicting uncertainty to be beneficial for training.

**Training:** We train our model using the ground-truth trajectories $\mathbf{p}_t \in \mathbb{R}^{N \times 2}$ and visibility information $\mathbf{v}_t \in \{0, 1\}^N$. For patch classification, we apply cross-entropy loss based on the ground-truth class, patch $\mathbf{c}^{patch}$. For offset prediction $\hat{\mathbf{o}}_t$, we minimize the $\ell_1$ distance between the predicted offset and the actual offset. We supervise the visibility $\hat{\mathbf{v}}_t$ and uncertainty $\hat{\mathbf{u}}_t$ using binary cross-entropy loss. Additionally, we supervise the uncertainties of the top-$k$ points, $\hat{\mathbf{u}}_t^{top}$, at re-ranking. The total loss is a weighted combination of them:

$$\mathcal{L} = \lambda \underbrace{\left( \mathcal{L}_{\text{CE}}\left( \mathbf{C}_t, \ \mathbf{c}^{patch} \right) + \mathcal{L}_{\text{CE}}\left( \mathbf{C}_t^{dec}, \ \mathbf{c}^{patch} \right) \right)}_{\text{Patch Classification Loss}} \cdot \mathbf{v}_t$$
$$+ \underbrace{\mathcal{L}_{\ell_1}\left( \hat{\mathbf{o}}_t, \ \mathbf{o}_t \right)}_{\text{Offset Loss}} \cdot \mathbf{v}_t + \underbrace{\mathcal{L}_{\text{CE}}(\hat{\mathbf{v}}_t, \mathbf{v}_t)}_{\text{Visibility Loss}} + \underbrace{\mathcal{L}_{\text{CE}}(\hat{\mathbf{u}}_t, \mathbf{u}_t)}_{\text{Uncertainty Loss}} + \underbrace{\mathcal{L}_{\text{CE}}(\hat{\mathbf{u}}_t^{top}, \mathbf{u}_t^{top})}_{\text{Top-$k$ Uncertainty Loss}} \qquad (6)$$

**Discussion**: Till this point, our model has exclusively considered relocating the queried points within the current frame. However, as the appearance of points consequently changes over time, the embedding similarity between the initial query point and future correspondences tends to decrease

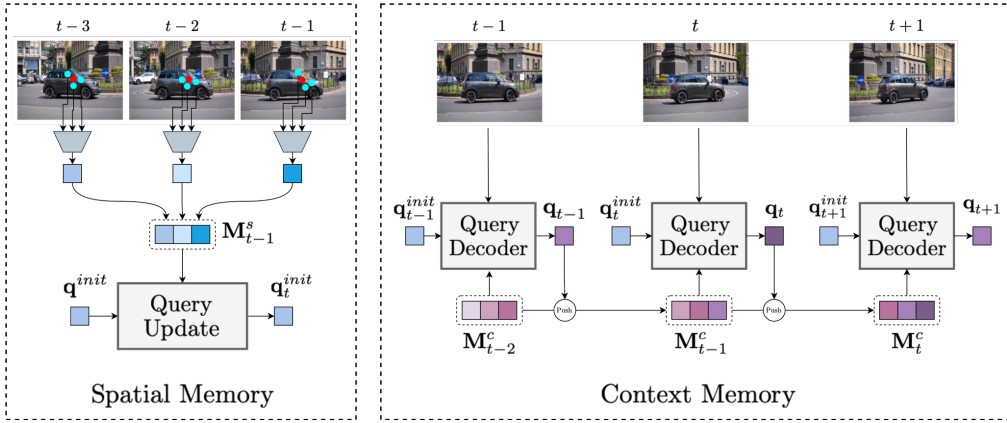

Figure 7: **Memory Modules.** Spatial memory $\mathbf{M}_{t-1}^s$ (**left**) is used to update the initial query $\mathbf{q}^{init}$ from the first frame to $\mathbf{q}_t^{init}$ on the current frame. The goal is to resolve feature drift by storing the content around the model's predictions in previous frames. Context memory $\mathbf{M}_{t-1}^c$ (**right**) is input to the query decoder which updates $\mathbf{q}_t^{init}$ to $\mathbf{q}_t$. It provides a broader view of the track's history with appearance changes and occlusion status by storing the point's embeddings from past frames.

gradually (Fig. 6). This problem, known as feature drift, leads to inaccurate predictions, when solely relying on the feature similarity with the initial point.

## 2.3 TRACK-ON WITH MEMORY

Here, we introduce two types of memories: **spatial memory** and **context memory**. Spatial memory stores information around the predicted locations, allowing us to update the initial queries based on the latest predictions. Context memory preserves the track's history states by storing previously decoded queries, ensuring continuity over time and preventing inconsistencies. This design enables our model to effectively capture temporal progressions in long-term videos, while also adapting to changes in the target's features to address feature drift.

We store the past features for each of the $N$ queries independently, with up to $K$ embeddings per query in each memory module. Once fully filled, the earliest entry from the memory will be obsoleted as a new entry arrives, operating as a First-In First-Out (FIFO) queue.

### 2.3.1 SPATIAL MEMORY

Here, we introduce the spatial memory module that stores fine-grained local information from previous frames, enabling continuous updates to the initial query points. This adaptation to appearance changes helps mitigate feature drift.

**Memory Construction:** We zero-initialize the memory, $\mathbf{M}_0^s$, update its content with each frame. For the first frame, we make a prediction using initial query $\mathbf{q}^{init}$ without memory.

**Memory Write ($\Phi_{\text{q-wr}}$):** To update the memory with the new prediction, $\mathbf{M}_{t-1}^s \rightarrow \mathbf{M}_t^s$, we extract a feature vector around the predicted point $\hat{\mathbf{p}}_t$ on the current feature map $\mathbf{f}_t$, and add it to the memory:

$$\mathbf{M}_t^s = [\mathbf{M}_{t-1}^s, \ \Phi_{\text{q-wr}}\left([\mathbf{q}^{init}, \mathbf{q}_t], \ \mathbf{f}_t, \ \hat{\mathbf{p}}_t\right)] \tag{7}$$

$\Phi_{\text{q-wr}}$ is a 3-layer deformable transformer decoder without self-attention, using the concatenated $\mathbf{q}^{init}$ and $\mathbf{q}_t$ as the query, attending a local neighborhood of predicted point for update. Utilizing deformable attention for the local summarization process helps prevent error propagation over time, as the query can flexibly select relevant features from any range.

**Query Update ($\Phi_{\text{q-up}}$):** In such scenario, before passing into the query decoder to estimate the correspondence, the initial query points first visit the spatial memory $\mathbf{M}_{t-1}^s$ for an update:

$$\mathbf{q}_t^{init} = \Phi_{\text{q-up}}\left(\mathbf{q}^{init}, \ \mathbf{M}_{t-1}^s\right) = \mathbf{q}^{init} + \phi_{\text{qqm}}\left(\mathbf{q}^{init}, \ \phi_{\text{mm}}(\mathbf{M}_{t-1}^s + \gamma^s)\right) \tag{8}$$

$\phi_{\text{mm}}$ is a transformer encoder layer that captures dependencies within the memory, and $\phi_{\text{qqm}}$ is a transformer decoder layer without initial self-attention, where $\mathbf{q}^{init}$ attends to updated memory, followed by a linear layer, and $\gamma^s \in \mathbb{R}^{K \times D}$ is learnable position embeddings. Instead of sequentially updating the query embeddings at each time step, *e.g.* extracting $\mathbf{q}_t^{init}$ using $\mathbf{q}_{t-1}^{init}$, we update them with respect to the initial query $\mathbf{q}^{init}$, conditioned on all previous predictions stored in the memory. This prevents error propagation by taking into account the entire history of predictions.

### 2.3.2 CONTEXT MEMORY

In addition to spatial memory, we introduce a context memory that incorporates historical information of the queried points from a broader context, enabling the model to capture past occlusions and visual changes. Specifically, we store the decoded query features from previous time steps in context memory, $\mathbf{M}_{t-1}^c$. We then integrate it by extending the query decoder with an additional transformer decoder layer without self-attention, where queries attend to memory with added learnable position embeddings ($\gamma^c \in \mathbb{R}^{K \times D}$):

$$\mathbf{q}_t = \Phi_{\text{q-dec}} \left( \mathbf{q}_t^{init}, \ \mathbf{f}_t, \ \mathbf{M}_{t-1}^c + \gamma^c \right) \tag{9}$$

Changes to the query decoder with memory are shown in red. For the writing operation, we add the most recent $\mathbf{q}_t$ to $\mathbf{M}_{t-1}^c$ and remove the oldest item, following the same procedure as in the spatial memory. Our experiments demonstrate that incorporating past content temporally with context memory enables more consistent tracking with additional benefits over spatial memory, especially in visibility prediction, since spatial memory focuses only on the regional content where the point is currently visible.

### 2.3.3 INFERENCE-TIME MEMORY EXTENSION

Although the memory size $K$ is fixed at training time, the number of video frames at inference can be different from the training frame limit. To address this, we extend the memory size during inference by linearly interpolating the temporal positional embeddings, $\gamma^s$ and $\gamma^c$, to a larger size $K_i$. In particular, we train our model with memory size $K = 12$, and extend it to $K_i \in \{16, \dots, 96\}$ at inference time.

## 3 EXPERIMENTS

### 3.1 EXPERIMENTAL SETUP

**Datasets:** We use TAP-Vid (Doersch et al., 2022) for both training and evaluation, consistent with previous work. Specifically, we train our model on TAP-Vid Kubric, a synthetic dataset of 11k video sequences, each with a fixed length of 24 frames. For evaluation, we use three other datasets from the TAP-Vid benchmark: TAP-Vid DAVIS, which includes 30 real-world videos from the DAVIS dataset; TAP-Vid RGB-Stacking, a synthetic dataset of 50 videos focused on robotic manipulation tasks, mainly involving textureless objects; TAP-Vid Kinetics, a collection of over 1,000 real-world online videos. We provide comparisons on four additional datasets in Appendix Sec. C.

**Metrics:** We evaluate tracking performance with the following metrics of TAP-Vid benchmark: Occlusion Accuracy (OA), which measures the accuracy of visibility prediction; $\delta_{avg}^x$, the average proportion of visible points tracked within 1, 2, 4, 8, and 16 pixels; Average Jaccard (AJ), which jointly assesses visibility and localization precision.

**Evaluation Details:** We follow the standard protocol of TAP-Vid benchmark by first downsampling the videos to $256 \times 256$. We evaluate models in the queried first protocol, which is the natural setting for causal tracking. In this mode, the first visible point in each trajectory serves as the query, and the goal is to track that point in subsequent frames. For DAVIS evaluation, we set the memory size $K_i$ to 48, 80, and 96 for DAVIS, RGB-Stacking, and Kinetics, to accommodate a larger temporal span.

### 3.2 RESULTS

As shown in Table 1, we categorize models into online and offline settings. Offline models, with bidirectional information flow, use either a fixed-size window—where half of the window spans past

Table 1: **Quantitative Results on TAP-Vid Benchmark.** This table shows results in comparison to the previous work on TAP-Vid under queried first setting, in terms of AJ, $\delta_{avg}^x$, and OA. The models are categorized into online and offline schemes, the former setting grants access to any frame regardless of video length, thus providing a clear advantage. While online models process one frame at a time, enable frame-by-frame inference. For training datasets, Kub and Kub-L(ong), refer to the TAP-Vid Kubric dataset with 24-frame and 64-frame videos, respectively; and R indicates the inclusion of a large number of real-world videos, we highlight these models in gray . MFT is a long-term optical flow method trained on a combination of Sintel (Butler et al., 2012), FlyingThings (Mayer et al., 2016), and Kubric datasets.

| Model | Input | Train | DAVIS | | | RGB-Stacking | | | Kinetics | | |
|---|---|---|---|---|---|---|---|---|---|---|---|
| | | | AJ ↑ | $\delta_{avg}^x$↑ | OA ↑ | AJ ↑ | $\delta_{avg}^x$↑ | OA ↑ | AJ ↑ | $\delta_{avg}^x$↑ | OA ↑ |
| **Offline** | | | | | | | | | | | |
| TAPIR | Video | Kub | 56.2 | 70.0 | 86.5 | 55.5 | 69.7 | 88.0 | 49.6 | 64.2 | 85.0 |
| TAPTR | Window | Kub | 63.0 | 76.1 | 91.1 | 60.8 | 76.2 | 87.0 | 49.0 | 64.4 | 85.2 |
| TAPTRv2 | Window | Kub | 63.5 | 75.9 | **91.4** | 53.4 | 70.5 | 81.2 | 49.7 | 64.2 | 85.7 |
| SpatialTracker | Window | Kub | 61.1 | 76.3 | 89.5 | 63.5 | 77.6 | 88.2 | 50.1 | 65.9 | 86.9 |
| LocoTrack | Video | Kub | 62.9 | 75.3 | 87.2 | 69.7 | 83.2 | 89.5 | 52.9 | 66.8 | 85.3 |
| CoTracker3 | Window | Kub-L | 64.5 | 76.7 | 89.7 | 71.1 | 81.9 | 90.3 | **54.1** | 66.6 | 87.1 |
| CoTracker3 | Video | Kub-L | 63.3 | 76.2 | 88.0 | **74.0** | 84.9 | 90.5 | 53.5 | 66.5 | 86.4 |
| BootsTAPIR | Video | Kub + R | 61.4 | 73.6 | 88.7 | 70.8 | 83.0 | 89.9 | 54.6 | 68.4 | 86.5 |
| CoTracker3 | Window | Kub-L + R | 63.8 | 76.3 | 90.2 | 71.7 | 83.6 | 91.1 | 55.8 | 68.5 | 88.3 |
| CoTracker3 | Video | Kub-L + R | 64.4 | 76.9 | 91.2 | 74.3 | 85.2 | 92.4 | 54.7 | 67.8 | 87.4 |
| **Online** | | | | | | | | | | | |
| DynOMo | Frame | - | 45.8 | 63.1 | 81.1 | - | - | - | - | - | - |
| MFT | Frame | SFK | 47.3 | 66.8 | 77.8 | - | - | - | 39.6 | 60.4 | 72.7 |
| Online TAPIR | Frame | Kub | 56.7 | 70.2 | 85.7 | 67.7 | - | - | 51.5 | 64.4 | 85.2 |
| Track-On (*Ours*) | Frame | Kub | **65.0** | **78.0** | 90.8 | 71.4 | 85.2 | 91.7 | 53.9 | 67.3 | **87.8** |

frames and the other half future frames—or the entire video, granting access to any frame regardless of video length and providing a clear advantage. In contrast, online models process one frame at a time, enabling frame-by-frame inference. In the following discussion, we mainly focus on the setting using similar training set to ours, *i.e.*, the models without using real-world videos.

**Comparison on DAVIS.** Our model outperforms all existing online models across every evaluation metric, achieving an 8.3 AJ improvement over the closest competitor, Online TAPIR. Additionally, it surpasses all offline models in both AJ (65.0 vs. 64.5) and $\delta_{avg}^x$ (78.0 vs. 76.7), outperforming even the concurrent CoTracker3, which was trained on longer videos (24 vs. 64 frames). Notably, our model also outperforms models fine-tuned on real-world videos by a significant margin. These results are particularly impressive because our model is an online approach, processing the video frame by frame, yet it exceeds the performance of offline models that process the entire video at once.

**Comparison on RGB-Stacking.** The dataset consists of long video sequences, with lengths of up to 250 frames, making it ideal for evaluating models' long-term processing capabilities. Our model surpasses Online TAPIR by 3.7 AJ and outperforms offline competitors, achieving improvements of 0.3 in $\delta_{avg}^x$ and 1.2 in OA compared to CoTracker3, which utilizes video-level input. The results of offline models on this dataset highlight a significant limitation of the windowed inference approach, which struggles with long video sequences due to restricted temporal coverage. In contrast, models with full video input perform considerably better. By effectively extending the temporal span through our memory mechanisms, our model achieves comparable or superior performance on long videos using only frame-by-frame inputs, despite the inherent disadvantage of not having bidirectional connections across the entire video sequence.

**Comparison on Kinetics.** The dataset comprises a variety of long internet videos. Our model outperforms Online TAPIR across all metrics by a considerable margin, while also surpassing offline models in $\delta_{avg}^x$ and OA. Specifically, it achieves a 0.5 improvement in $\delta_{avg}^x$ over LocoTrack (with video inputs) and a 0.7 improvement in OA over CoTracker3 (with window inputs). Despite the significant difference in training data between CoTracker3 and our model, ours ranks second in AJ, with only a small gap of 0.2. Additionally, models fine-tuned on real-world data demonstrate superior

Table 2: **Model Components.** Removing individual components of our model without (inference-time memory extension)—namely, the re-ranking module ($\Phi_{\text{rank}}$), offset head ($\Phi_{\text{off}}$), and visibility head ($\Phi_{\text{vis}}$) one at a time. All metrics are higher-is-better.

| Model | $\delta^{1px}$ | $\delta^{16px}$ | AJ | $\delta^x_{avg}$ | OA |
|---|---|---|---|---|---|
| Full Model (without IME) | 45.5 | 95.9 | 64.9 | 77.7 | 90.6 |
| - No re-ranking ($\Phi_{\text{re-rank}}$) | 43.8 | 95.5 | 62.8 | 76.3 | 89.7 |
| - No offset head ($\Phi_{\text{off}}$) | 27.6 | 96.1 | 60.1 | 73.0 | 90.5 |
| - No visibility head ($\Phi_{\text{vis}}$) | 45.4 | 96.1 | 64.0 | 77.4 | 90.6 |

Table 3: **Memory Components.** The effect of spatial memory ($\mathbf{M}^s$), context memory ($\mathbf{M}^c$), and inference-time memory extension (IME). All metrics are higher-is-better.

| Model | $\mathbf{M}^s$ | $\mathbf{M}^c$ | IME | AJ | $\delta^x_{avg}$ | OA |
|---|---|---|---|---|---|---|
| A | ✗ | ✗ | ✗ | 52.0 | 67.6 | 78.1 |
| B | ✓ | ✗ | ✗ | 63.5 | 77.0 | 89.0 |
| C | ✗ | ✓ | ✗ | 64.3 | 77.8 | 90.3 |
| D | ✓ | ✓ | ✗ | 64.9 | 77.7 | 90.6 |
| E | ✓ | ✓ | ✓ | 65.0 | 78.0 | 90.8 |

performance, underscoring the potential benefits of training on large-scale real-world datasets, which seem particularly advantageous for datasets like Kinetics compared to others.

## 3.3 ABLATION STUDY

**Components:** We conducted an experiment to examine the impact of each proposed component in the correspondence estimation section (Sec. 2.2), we remove them one at a time while keeping other modules unchanged. First, we removed the re-ranking module $\Phi_{\text{rank}}$. Second, we removed the offset head $\Phi_{\text{off}}$, eliminating the calculation of additional offsets. Instead, we used the coarse prediction, *i.e.* the selected patch center, as the final prediction. A more detailed analysis of the offset head is provided in Appendix Sec. D. Lastly, we replaced the additional deformable attention layer in the visibility head $\Phi_{\text{vis}}$ with a 2-layer MLP. Note that, we do not apply inference-time memory extension to models in this comparison.

From the results in Table 2, we can make the following observations: (i) The re-ranking module improves all metrics, notably increasing AJ by 2.1, as it introduces specialized queries for identifying correspondences. Errors larger than 16 pixels are also more frequent without it, showing its role in reducing large errors. (ii) The offset head is crucial for fine-grained predictions. While $\delta^{16px}$ values remain similar without the offset head, lower error thresholds (*i.e.* less than 1 pixel) show a significant difference (45.5 vs. 27.6), highlighting the importance of predicted offsets for fine-grained localization. (iii) Replacing the deformable attention layer in $\Phi_{\text{vis}}$ with an MLP does not affect OA but reduces AJ. The deformable head ensures more consistent visibility predictions by conditioning them on accurate point predictions, leading to higher AJ. Despite this, OA remains robust even when an MLP is used for visibility prediction.

**Memory Modules:** To demonstrate the effectiveness of our proposed memory modules, we conduct an ablation study, as shown in Table 3. We start by evaluating the model without memory (Model-A), which corresponds to the vanilla model described in Sec. 2.2. As expected, due to the model's lack of temporal processing, Model-A performs poorly, particularly in OA. Introducing temporal information through either spatial memory (Model-B) or context memory (Model-C) leads to significant performance improvements. Model-C, in particular, achieves higher OA by providing a more comprehensive view of the track's history, including occlusions. Combining both memory types (Model-D) further boosts performance, highlighting the complementary strengths of the two memory modules. Lastly, incorporating the memory extension at inference time yields slight improvements in all metrics, leading to an overall enhancement in performance. We provide more detailed analysis on spatial memory in Appendix Sec. E.

**Efficiency:** We plot the inference speed (frames per second, FPS), maximum GPU memory usage during video processing, and AJ performance on the TAP-Vid DAVIS dataset as a function of memory size $K_i$ (indicated near the plot nodes) in Fig. 8. The results are based on tracking approximately 400 points on a single NVIDIA A100 GPU. Unlike offline methods, our approach does not utilize temporal parallelization in the visual encoder, processing frames sequentially in an online setting. As the memory size $K$ increases, the model's inference speed decreases due to the higher computational cost of temporal attention in memory operations, correspondingly increasing GPU memory usage. For instance, the FPS decreases from 19.2 with $K = 12$ to 16.8 with $K = 48$, and further down to 14.1 with $K = 96$.

Additionally, our model demonstrates high memory efficiency, with GPU memory usage ranging from 0.61 GB ($K = 12$) to a maximum of 1.03 GB ($K = 96$). At the default memory size of $K = 48$, where our model performs best on this dataset, it achieves 16.8 FPS with a maximum GPU memory usage of 0.73 GB. This highlights the efficiency of our frame-by-frame tracking approach, making it well-suited for consumer GPUs and real-time applications. Moreover, we observe that performance improves as the memory size increases up to $K = 48$, but declines beyond this point. This suggests that excessively large memory sizes can hurt performance by storing unnecessary information. Additional analysis of memory size is provided in Appendix Sec. D.

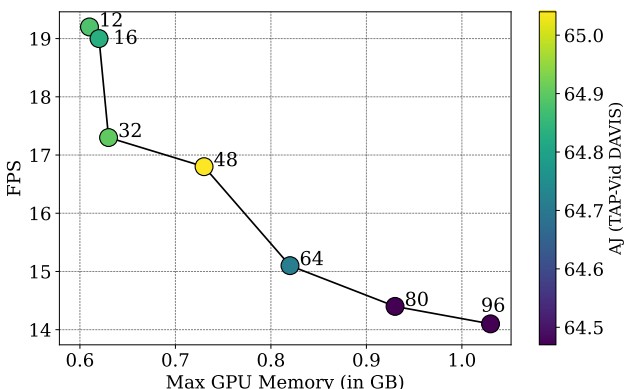

Figure 8: **Efficiency.** Inference speed (frames per second, FPS) vs. maximum GPU memory usage (in GB) where color represents the performance in AJ for different memory sizes (indicated near the nodes), while tracking approximately 400 points on the DAVIS dataset.

## 4 RELATED WORK

**Tracking Any Point:** Point tracking presents significant challenges, particularly in long-term scenarios involving occlusions and appearance changes. Early methods like PIPs (Harley et al., 2022) relied on iterative updates, while TAPIR (Doersch et al., 2023) focused on refining initialization and improving temporal accuracy. CoTracker (Karaev et al., 2024b) leveraged spatial correlations to jointly track multiple points, and TAPTR (Li et al., 2024b) adopted a DETR-inspired design for tracking. More recent approaches, such as LocoTrack (Cho et al., 2024) and CoTracker3 (Karaev et al., 2024a), introduced region-to-region similarity for enhanced matching and utilized pseudo-labeled data to boost performance. However, most of these methods operate offline, requiring access to entire video frames or fixed windows. In contrast, our approach focuses on online tracking, employing memory modules to effectively capture temporal information. Additionally, we diverge from regression-based iterative updates, instead adopting a patch classification and refinement paradigm.

**Causal Processing in Videos:** Online models process frames sequentially, without access to future frames, making them well-suited for streaming and real-time tasks (Xu et al., 2021; Zhou et al., 2024). This has been explored in tasks like pose estimation (Nie et al., 2019), action detection (Wang et al., 2021), and video segmentation (Cheng & Schwing, 2022). To enhance efficiency, approaches such as XMem (Cheng & Schwing, 2022) and LSTR (Xu et al., 2021) incorporate memory modules to balance long-term and short-term contexts. Similarly, we employ an attention-based memory mechanism tailored for point tracking, with spatial and contextual memories for capturing both local and global information.

## 5 CONCLUSION & LIMITATION

In this work, we presented **Track-On**, a simple yet effective transformer-based model for online point tracking. To establish correspondences, our model employs patch classification, followed by further refinement with offset prediction. We proposed two memory modules that enable temporal continuity efficiently while processing long videos. Our model significantly advances the state-of-the-art in online point tracking with fast inference and narrows the performance gap between online and offline models across a variety of public datasets.

Despite the strengths of our proposed model, there remain certain limitations. Specifically, the model may suffer from precision loss on thin surfaces and struggle to distinguish between instances with similar appearances, as observed in our failure cases (see Appendix). Future work could address these challenges by exploring learnable upsampling techniques to achieve higher-resolution feature maps and improve feature sampling accuracy.

## 6 ACKNOWLEDGEMENTS

We would like to thank Shadi Hamdan and Merve Rabia Barın for their remarks and assistance. This project is funded by the European Union (ERC, ENSURE, 101116486) with additional compute support from Leonardo Booster (EuroHPC Joint Undertaking, EHPC-AI-2024A01-060). Views and opinions expressed are however those of the author(s) only and do not necessarily reflect those of the European Union or the European Research Council. Neither the European Union nor the granting authority can be held responsible for them. Weidi Xie would like to acknowledge the National Key R&D Program of China (No. 2022ZD0161400).

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

# Appendices

## A EXTENDED RELATED WORK

**Tracking Any Point:** Point tracking, presents significant challenges, particularly for long-term tracking where maintaining consistent tracking through occlusions is difficult. PIPs (Harley et al., 2022) was one of the first approaches to address this by predicting motion through iterative updates within temporal windows. TAP-Vid (Doersch et al., 2022) initiated a benchmark for evaluation. TAPIR (Doersch et al., 2023) improved upon PIPs by refining initialization and incorporating depthwise convolutions to enhance temporal accuracy. BootsTAPIR (Doersch et al., 2024) further advanced TAPIR by utilizing student-teacher distillation on a large corpus of real-world videos. In contrast, CoTracker (Karaev et al., 2024b) introduced a novel approach by jointly tracking multiple points, exploiting spatial correlations between points via factorized transformers. Differently, TAPTR (Li et al., 2024b) adopted a design inspired by DETR (Carion et al., 2020; Zhu et al., 2021), drawing parallels between object detection and point tracking. DINO-Tracker (Tumanyan et al., 2024) took a different route, using DINO as a foundation for test-time optimization, whose tracking capabilities have been shown (Aydemir et al., 2024) to be one of the best among foundation models. TAPTRv2 (Li et al., 2024a), the successor to TAPTR, builds on its predecessor by incorporating offsets predicted by the deformable attention module. While these models calculate point-to-region similarity for correlation, LocoTrack (Cho et al., 2024) introduced a region-to-region similarity approach to address ambiguities in matching. Recently, CoTracker3 (Karaev et al., 2024a) combined the region-to-region similarity method from LocoTrack with the original CoTracker architecture and utilized pseudo-labeled real-world data during training to further enhance performance.

However, all these models are designed for offline tracking, assuming access to all frames within a sliding window (Karaev et al., 2024b) or the entire video (Doersch et al., 2023; 2024). Conversely, MFT (Neoral et al., 2024), which extends optical flow to long-term scenarios, can be adapted for online point tracking tasks, although it does not belong to the point tracking family. Among point tracking approaches, models with online variants (Doersch et al., 2024; 2023) are re-trained with a temporally causal mask to process frames sequentially on a frame-by-frame basis, despite being originally designed for offline tracking. In contrast, we explicitly focus on online point tracking by design, enabled by novel memory modules to capture temporal information. Additionally, many of these models use a regression objective, originally developed for optical flow (Teed & Deng, 2020), while we introduce a new paradigm based on patch classification and refinement.

Another line of research, orthogonal to ours, explores leveraging scene geometry for point tracking. SpatialTracker (Xiao et al., 2024) extends CoTracker to the 3D domain by tracking points in three-dimensional space, while OmniMotion (Wang et al., 2023) employs test-time optimization to learn a canonical representation of the scene. Concurrent work DynOMO (Seidenschwarz et al., 2025) also uses test-time optimization, utilizing Gaussian splats for online point tracking.

**Causal Processing in Videos:** Online, or temporally causal models rely solely on current and past frames without assuming access to future frames. This is in contrast to current practice in point tracking with clip-based models, processing frames together. Causal models are particularly advantageous for streaming video understanding (Yang et al., 2022a; Zhou et al., 2024), embodied perception (Yao et al., 2019), and processing long videos (Zhang et al., 2024; Xu et al., 2021), as they process frames sequentially, making them well-suited for activation caching. Due to its potential, online processing has been studied across various tasks in computer vision, such as pose estimation (Fan et al., 2021; Nie et al., 2019), action detection (Xu et al., 2019; De Geest et al., 2016; Kondratyuk et al., 2021; Eun et al., 2020; Yang et al., 2022b; Wang et al., 2021; Zhao & Krähenbühl, 2022; Xu et al., 2021; Chen et al., 2022a), temporal action localization (Buch et al., 2017; Singh et al., 2017), object tracking (He et al., 2018; Wang et al., 2020), video captioning (Zhou et al., 2024), and video object segmentation (Cheng & Schwing, 2022; Liang et al., 2020).

In causal models, information from past context is commonly propagated using either sequential models (De Geest et al., 2016), which are inherently causal, or transformers with causal attention masks (Wang et al., 2021). However, these models often struggle to retain information over long contexts or face expanded memory requirements when handling extended past contexts. To address this, some approaches introduce memory modules for more effective and efficient handling of

complex tasks. For example, LSTR (Xu et al., 2021) separates past context as long-term and short-term memories for action detection, while XMem (Cheng & Schwing, 2022) incorporates a sensory memory module for fine-grained information in video object segmentation. Long-term memory-based modeling is also applied beyond video understanding (Balazevic et al., 2024), including tasks like long-sequence text processing and video question answering (Zhang et al., 2021). We also employ an attention-based memory mechanism, which is specialized in point tracking with two types of memory; one focusing on spatial local regions around points, and another on broader context.

## B EXPERIMENT DETAILS

### B.1 TRAINING DETAILS

We train our model for 150 epochs, equivalent to approximately 50K iterations, using a batch size of 32. The model is optimized using the AdamW optimizer (Loshchilov & Hutter, 2019) on $32 \times$ A100 64GB GPUs, with mixed precision. The learning rate is set to a maximum of $5 \times 10^{-4}$, following a cosine decay schedule with a linear warmup period covering 5% of the total training time. A weight decay of $1 \times 10^{-5}$ is applied, and gradient norms are clipped at 1.0 to ensure stable training. Input frames are resized to $384 \times 512$ using bilinear interpolation before processing.

For training, we utilize entire clips of length 24 from TAP-Vid Kubric. We adopt the data augmentation techniques from CoTracker (Karaev et al., 2024b), including random cropping to a size of $384 \times 512$ from the original $512 \times 512$ frames, followed by random Color Jitter and Gaussian Blur. Each training sample includes up to $N = 480$ points. We apply random key masking with a 0.1 ratio during attention calculations for memory read operations throughout training.

For the training loss coefficients, we set $\lambda$ to 3. During training, we clip the offset loss to the stride $S$ to prevent large errors from incorrect patch classifications and stabilize the loss. Deep supervision is applied to offset head ($\Phi_{\text{off}}$), and the average loss across layers is used. We set the softmax temperature $\tau$ to 0.05 in patch classification. We set the visibility threshold to 0.8 for all datasets except RGB-Stacking, where it is set to 0.5 due to its domain-specific characteristics, consisting of simple, synthetic videos.

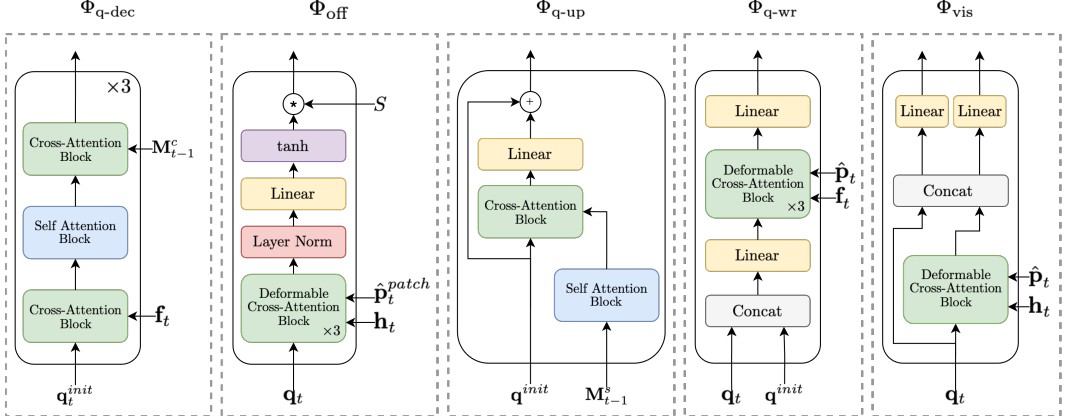

Figure 9: **Details of Different Modules**. This figure describes the details for modules in our model: Query Decoder ($\Phi_{\text{q-dec}}$), Offset Head ($\Phi_{\text{off}}$), Query Update Module ($\Phi_{\text{q-up}}$), Spatial Memory Write Module ($\Phi_{\text{q-wr}}$), and Visibility Head ($\Phi_{\text{vis}}$).

### B.2 IMPLEMENTATION DETAILS

All of our modules consist of either a Self-Attention Block, Cross-Attention Block, or Deformable Cross-Attention Block. Each block includes multi-head self-attention, multi-head cross-attention, or multi-head deformable cross-attention, followed by a 2-layer feed-forward network with a hidden dimension expansion ratio of 4. Each multi-head attention uses 8 heads, while deformable multi-head attention operates with 4 levels, sampling 4 points per head. We extract multi-level feature

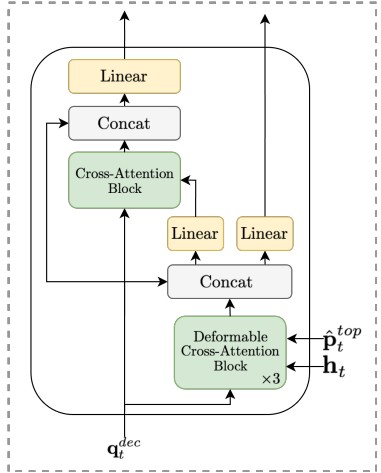

Figure 10: **Re-ranking Module.** This figure describes the detailed architecture of re-ranking module ($\Phi_{\text{rank}}$).

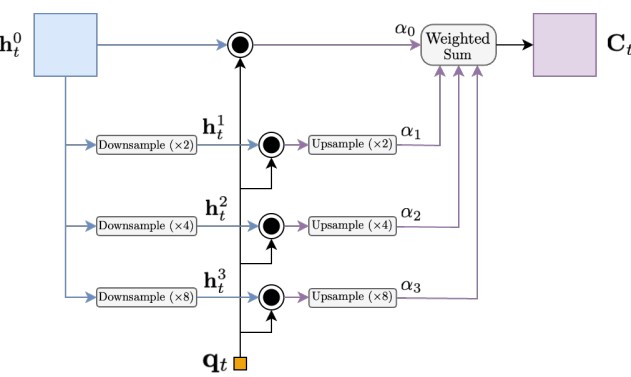

Figure 11: **Multiscale Similarity Calculation.** This figure illustrates the detailed process of computing multiscale similarity between a given query $\mathbf{q}_t$ and a feature map $\mathbf{h}_t$. The different levels of the feature map ($\mathbf{h}_t^l$) are generated by applying bilinear downsampling at various scales.

maps by downsampling the input feature map and set the feature dimension $D$ to 256. Following CoTracker (Karaev et al., 2024b), we add a global support grid of size $20 \times 20$ during inference.

**Visual Encoder** ($\Phi_{\text{vis-enc}}$): We use the ViT-Adapter (Chen et al., 2022b) with DINOv2 ViT-S/14 (Oquab et al., 2024; Käppeler et al., 2024) as the backbone. The DINOv2 inputs are resized to $378 \times 504$, as the default input size of $384 \times 512$ is not divisible by the patch size of 14. The backbone outputs, with a dimension of 384, are projected to $D$ using a single linear layer.

**Query Decoder** ($\Phi_{\text{q-dec}}$): Query Decoder is shown in Fig. 9, first block. We set the number of layers to 3. Positional embedding $\gamma^c$ from the context memory is applied only to the keys, not the values, ensuring time-invariance in the queries while enabling the model to differentiate between time steps during attention score calculation.

**Offset Prediction** ($\Phi_{\text{off}}$): Second block in Fig. 9 shows the architecture of the offset prediction head. We set the number of layers to 3. Following DETR (Carion et al., 2020), we normalize the queries before projecting them through a linear layer, as the per-layer loss is calculated for the offset head.

**Query Update** ($\Phi_{\text{q-up}}$): The query update is detailed in third block of Fig. 9. In both attention blocks, we mask items corresponding to frames where points are predicted as occluded.

**Spatial Memory Write** ($\Phi_{\text{q-wr}}$): The spatial memory writer module is depicted in the fourth block of Fig. 9. We set the number of layer to 3.

**Visibility Head** ($\Phi_{\text{vis}}$): The visibility head is shown in the last block of Fig. 9. We decode the query feature around the predicted point and concatenate it with the input query. Two separate linear layers are then applied to predict visibility and uncertainty.

**Re-ranking Module** ($\Phi_{\text{re-rank}}$): The re-ranking module is detailed in Fig. 10. Initially, the given query $\mathbf{q}_t^{dec}$ is decoded around the top-$k$ points using a Deformable Cross-Attention block with 3 layers. The resulting features are concatenated with the input query feature to directly incorporate information from earlier stages. These correspond to the top-$k$ features, *i.e.* $\mathbf{q}_t^{top} \in \mathbb{R}^{N \times k \times D}$. Next, these features are fused into the input query through a single Cross-Attention block, followed by a separate linear layer. The output is concatenated with the input once more, and a final linear transformation reduces the dimensionality from $2D$ to $D$ (*left upper arrow*). Additionally, the uncertainties for these top-$k$ locations are predicted directly from $\mathbf{q}_t^{top}$ using a linear layer (*right upper arrow*).

**Multiscale Similarity Calculation:** The multiscale similarity calculation used in patch classification (Section 2.2.3) is detailed in Fig. 11. Starting with a feature map $\mathbf{h}_t \in \mathbb{R}^{\frac{H}{S} \times \frac{W}{S} \times D}$, we generate 4-scale representations by applying bilinear downsampling, resulting in $\mathbf{h}_t^l \in \mathbb{R}^{\frac{H}{2^l \cdot S} \times \frac{W}{2^l \cdot S} \times D}$. For each scale, cosine similarity is calculated between the feature maps and any query feature $\mathbf{q}_t$, producing similarity maps at the respective scales. These maps are then upsampled back to the resolution of

Table 4: **Quantitative Results on RoboTAP, Dynamic Replica, and BADJA** This table shows results in comparison to the previous work on RoboTAP, Dynamic Replica, and BADJA under queried first setting. Similar to the main paper, the models are categorized into online and offline schemes, the former setting grants access to any frame regardless of video length, thus providing a clear advantage. While online models process one frame at a time, enable frame-by-frame inference. For training datasets, Kub and Kub-L(ong), refer to the TAP-Vid Kubric dataset with 24-frame and 64-frame videos, respectively; and R indicates the inclusion of a large number of real-world videos, we highlight these models in gray .

| Model | Input | Train | RoboTAP | | | Dynamic Replica | BADJA | |
|---|---|---|---|---|---|---|---|---|
| | | | AJ $\uparrow$ | $\delta^x_{avg}\uparrow$ | OA $\uparrow$ | $\delta^{vis}\uparrow$ | $\delta^{seg}\uparrow$ | $\delta^{3px}\uparrow$ |
| **Offline** | | | | | | | | |
| TAPIR | Video | Kub | 59.6 | 73.4 | 87.0 | 66.1 | 66.9 | 15.2 |
| TAPTR | Window | Kub | 60.1 | 75.3 | 86.9 | 69.5 | 64.0 | 18.2 |
| TAPTRv2 | Window | Kub | 60.9 | 74.6 | 87.7 | - | - | - |
| SpatialTracker | Window | Kub | - | - | - | - | 69.2 | 17.1 |
| LocoTrack | Video | Kub | 62.3 | 76.2 | 87.1 | 71.4 | - | - |
| CoTracker3 | Window | Kub-L | 60.8 | 73.7 | 87.1 | 72.9 | - | - |
| CoTracker3 | Video | Kub-L | 59.9 | 73.4 | 87.1 | 69.8 | - | - |
| BootsTAPIR | Video | Kub + R | 64.9 | 80.1 | 86.3 | 69.0 | - | - |
| CoTracker3 | Window | Kub-L + R | 66.4 | 78.8 | 90.8 | 73.3 | - | - |
| CoTracker3 | Video | Kub-L + R | 64.7 | 78.0 | 89.4 | 72.2 | - | - |
| **Online** | | | | | | | | |
| Online TAPIR | Frame | Kub | 59.1 | - | - | - | - | - |
| Track-On (*Ours*) | Frame | Kub | **63.5** | **76.4** | **89.4** | **73.6** | **71.9** | **20.2** |

the input feature map, $\frac{H}{S} \times \frac{W}{S} \times D$, using bilinear upsampling. Finally, we compute a weighted summation of the upsampled maps across all scales using learned coefficients $\alpha_l$. The weighted summation is implemented as a $1 \times 1$ convolution without bias.

## C  ADDITIONAL COMPARISONS

Similar to the main paper, we mainly focus on the setting using similar training set to ours, *i.e.*, the models without using real-world videos.

**RoboTAP:**  We evaluate our model on the RoboTAP dataset (Vecerik et al., 2023), which consists of 265 real-world robotic sequences with an average length of over 250 frames, as shown in Table 4. We use the same metrics as the TAP-Vid benchmark: AJ, $\delta_{avg}$, and OA, with the memory size $K_i$ set to 48. Our model consistently surpasses existing online and offline models across all metrics. Specifically, in AJ and $\delta_{avg}$, our model outperforms the closest competitor, LocoTrack (which processes the entire video), by 1.2 and 0.2 points, respectively. Additionally, it exceeds the nearest competitor (TAPTRv2) in OA by 1.7 points. This demonstrates that our causal memory modules, which enable online tracking, are capable of effectively capturing the dynamics of long video sequences despite lacking bidirectional information flow across all frames. It is worth noting that this dataset, which features textureless objects, presents a significant challenge. Fine-tuning on real-world videos provides substantial improvements, as learning to track points on textureless objects is particularly difficult, as highlighted by models tuned on real-world datasets.

**Dynamic Replica:**  We compare to previous work on the Dynamic Replica dataset (Karaev et al., 2023), a benchmark designed for 3D reconstruction with 20 sequences, each consisting of 300 frames, as shown in Table 4. Following prior work (Karaev et al., 2024b), we evaluate models using $\delta^{vis}$, consistent with the TAP-Vid benchmark. Unlike previous work, we do not report $\delta^{occ}$, as our model is not supervised for occluded points. The memory size is set to $K_i = 48$. Despite being an online model, our model outperforms offline competitors, including those trained on longer sequences (CoTracker3, 73.6 vs. 72.9) and versions fine-tuned on real-world videos (73.6 vs. 73.3). This highlights the robustness of our model, particularly in handling longer video sequences effectively.

**BADJA:**  We compare to previous work on the BADJA challenge (Biggs et al., 2019), a dataset for animal joint tracking comprising 7 sequences, as shown in Table 4. Two metrics are used

Table 5: **Quantitative Results on PointOdyssey.** This table shows results in comparison to the previous work on PointOdyssey under queried first setting.

| Model | Input | Train | PointOdyssey | | | |
|---|---|---|---|---|---|---|
| | | | $\delta_{avg}^{vis}$ ↑ | $\delta_{avg}^{all}$ ↑ | MTE ↓ | Survival ↑ |
| TAP-Net | Frame | Kub | - | 23.8 | 92.0 | 17.0 |
| TAP-Net | Frame | PO | - | 28.4 | 63.5 | 18.3 |
| PIPs | Window | Kub | - | 16.5 | 147.5 | 32.9 |
| PIPs | Window | PO | - | 27.3 | 64.0 | 42.3 |
| PIPs++ | Window | PO | 32.4 | 29.0 | - | 47.0 |
| CoTracker | Window | PO | 32.7 | 30.2 | - | **55.2** |
| Track-On (*Ours*) | Frame | Kub | **38.1** | **34.2** | **28.8** | 49.5 |

for evaluation: $\delta^{seg}$, which measures the proportion of points within a threshold relative to the segmentation mask size (specifically, points within $0.2\sqrt{A}$, where $A$ is the area of the mask); and $\delta^{3px}$, the ratio of points tracked within a 3-pixel range. Given the dataset's low FPS nature, we kept the memory size at the original value of 12. Our model achieves state-of-the-art results by a significant margin, with a 2.7-point improvement in $\delta^{seg}$ over SpatialTracker and a 2.0-point improvement in $\delta^{3px}$ over TAPTR. These results highlight the flexibility of our inference-time memory extension, enabling the model to adapt effectively to data with varying characteristics.

**PointOdyssey:** We evaluated our model, trained on TAP-Vid Kubric, on the Point Odyssey (PO) (Zheng et al., 2023) dataset, which consists of 12 long videos with thousands of frames (up to 4325). The results are presented in Table 5. We adopted four evaluation metrics proposed in Point Odyssey: $\delta_{avg}^{vis}$, which measures the $\delta_{avg}$ metric from the TAP-Vid benchmark for visible points; $\delta_{avg}^{all}$, which calculates $\delta_{avg}$ for all points, including both visible and occluded ones; MTE (Median Trajectory Error), computed for all points; and Survival Rate, defined as the average number of frames until tracking failure (set to 50 pixels). The memory size $K_i$ was set to 96. From the results, we observe that PIPs trained on Kubric achieves a $\delta_{avg}^{vis}$ of 16.5, while the same model trained on PO with a larger window size achieves 27.3 ($\sim 65\%$ improvement). Notably, CoTracker does not report the performance of its model trained on Kubric but instead reports results for a model trained with sequences of length 56 on PO. These findings highlight the importance of training on PO to achieve higher performance across models. Our model, trained on Kubric, outperforms CoTracker and PIPs++ trained on PO in both $\delta_{avg}^{vis}$ and $\delta_{avg}^{all}$. Interestingly, while training on PO is critical for other models to achieve strong performance, our model demonstrates robustness by surpassing them even when trained on a different data distribution. Moreover, despite not being explicitly supervised for occluded points, our model still achieves superior $\delta_{avg}^{all}$. In terms of the Survival Rate, our model falls behind CoTracker trained on PO, despite its superior $\delta$ metrics. This further emphasizes the importance of training on PO to excel in this specific metric.

# D    ADDITIONAL ABLATIONS

**Memory Size:** We experimented with varying memory sizes, trained with $K = 12$, and extended them to different values $K = \{16, 32, 48, 64, 80, 96\}$ during inference on TAP-Vid DAVIS and TAP-Vid RGB-Stacking, as shown in Fig. 12. The plot shows the change in AJ compared to the default training memory size of 12 after extension. Memory sizes reported in Table 1 are marked with crosses. For DAVIS (blue), performance slightly increases up to a memory size of 48 (64.88 AJ → 65.01 AJ) but declines beyond that, indicating that excessive memory can negatively impact the model. In contrast, for RGB-Stacking (orange), memory size plays a more critical role due to the disparity in video frame counts between training (24 frames) and inference (250 frames), as well as the high FPS nature of the dataset. Performance consistently improves up to $K = 80$, yielding a 2.9 AJ increase. These results highlight that, although the model is trained with a fixed and relatively small memory size, extending memory during inference is possible to adapt the varying characteristics of different datasets.

**Top-$k$ points:** We experimented with varying the number of $k$ points used in re-ranking (Sec. 2.2.3) on TAP-Vid DAVIS, as shown in Fig. 13. Although the model is trained with a fixed top-$k$ value of

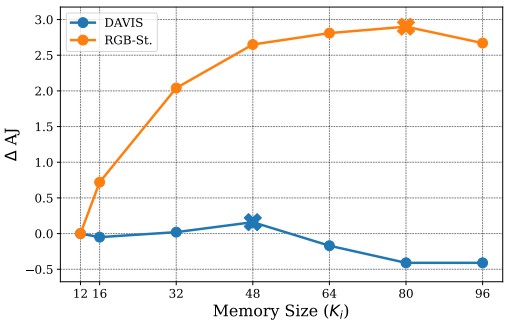 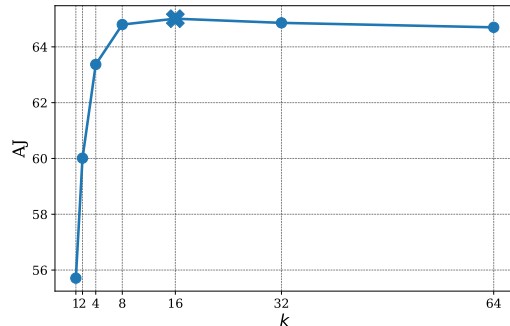

Figure 12: **Memory Size.** The effect of varying extended memory sizes during inference, on TAP-Vid DAVIS and TAP-Vid RGB-Stacking.

Figure 13: **Top-$k$ points.** The effect of varying $k$ number of top points in ranking module, during inference.

$k = 16$, this number can be adjusted during inference since the selected points are fused into the target query via a transformer decoder. The $k$ value reported in Table 1 is marked with cross. The results indicate that increasing the top-$k$ points up to 16 consistently improves performance. Beyond 16, performance decreases slightly but remains robust, even with larger $k$ values. This demonstrates that choosing a $k$ value smaller than the optimal number leads to a noticeable performance drop, emphasizing the importance of appropriately selecting $k$.

**Offset Head and Stride:** The offset head is essential for refining patch classification outputs, enabling more precise localization. Specifically, the offset head allows for precision beyond the patch size $S$ (stride). In Table 6, we examine the impact of removing the offset head ($\Phi_{\text{off}}$) for two stride values, $S = 4$ and $S = 8$, without utilizing inference-time memory extension. For both values, the addition of the offset head significantly enhances AJ and $\delta^x_{avg}$ by refining predictions within the local region. With stride 4, the offset head notably improves $\delta^{2px}$, while for stride 8, it improves both $\delta^{2px}$ and $\delta^{4px}$. This demonstrates that while patch classification offers coarse localization, the offset head provides further refinement, achieving pixel-level precision.

Larger stride values risk losing important details necessary for accurate tracking. For instance, increasing the stride from 4 to 8 results in AJ drops of 12% for TAPIR and 16% for CoTracker, as reported in their ablation studies. However, our coarse-to-fine approach mitigates the negative effects of stride 8, leading to only a minimal decline of 4%, highlighting the robustness of our model to larger stride values.

Note that the model with $S = 8$ and no offset head (first row) has a higher occlusion accuracy (OA). A possible reason is the imbalance in the loss, where the visibility loss has a relatively higher impact compared to the model with an additional offset loss (second row), leading to improved occlusion accuracy.

Table 6: **Offset Head.** The effect of removing the offset head ($\Phi_{\text{off}}$) on models with varying strides. All metrics are higher-is-better.

| $\Phi_{\text{off}}$ | Stride | $\delta^{2px}$ | $\delta^{4px}$ | $\delta^{8px}$ | AJ | $\delta^x_{avg}$ | OA |
|---|---|---|---|---|---|---|---|
| ✗ | 8 | 37.4 | 79.0 | 91.1 | 51.3 | 62.9 | 91.0 |
| ✓ | | 66.1 | 84.0 | 91.7 | 62.5 | 75.8 | 90.6 |
| ✗ | 4 | 64.3 | 84.4 | 92.4 | 60.1 | 73.0 | 90.5 |
| ✓ | | 69.3 | 85.5 | 92.5 | 64.9 | 77.7 | 90.6 |

# E   ANALYSIS OF SPATIAL MEMORY

To evaluate the effect of spatial memory in the presence of feature drift and inference-time memory extension, we conduct an experiment across different datasets using a model trained without spatial memory (Model-C in Table 3), as shown in Table 7. The results indicate that spatial memory

Table 7: **Spatial Memory.** Comparison of the model's performance with and without spatial memory ($\mathbf{M}^s$), evaluated using the AJ metric across different datasets, with inference-time memory extension (IME) applied.

| Model | DAVIS | RGB-Stacking | Kinetics | RoboTAP |
|---|---|---|---|---|
| Full Model | 65.0 | 71.4 | 53.9 | 63.5 |
| - Without Spatial Memory ($\mathbf{M}^s$) | 64.6 | 70.2 | 53.3 | 62.1 |

consistently improves AJ across four datasets: DAVIS, RGB-Stacking, Kinetics, and RoboTAP. The impact is particularly notable for RGB-Stacking (+1.2 AJ) and RoboTAP (+1.4 AJ), where objects are less descriptive and often textureless, as both datasets originate from robotics scenarios. This suggests that spatial memory, which retains information around the local region of previous predictions, helps mitigate drift and enhances generalization across different scene characteristics.

Additionally, to directly assess the impact of spatial memory (Sec. 2.3) in mitigating feature drift, we conducted an analysis comparing the tracking performance of the initial feature sampled from the query frame, $\mathbf{q}^{init}$, with the query feature updated using spatial memory at frame $t$, denoted as $\mathbf{q}_t^{init}$. For this evaluation, we introduced the new metric of similarity ratio score ($s_{sr}$), which measures how well the updated query features align with the feature at the target point compared to the initial query.

Ideally, $\mathbf{q}_t^{init}$ should provide a better starting point for detecting correspondences compared to $\mathbf{q}^{init}$, particularly when the object's appearance changes significantly. To assess whether $\mathbf{q}_t^{init}$ is more similar to the feature at the ground-truth correspondence location than $\mathbf{q}^{init}$, we calculate the ratio of their similarity to ground-truth, as a way of quantifying the increase in the similarity after the update:

$$s_{sr}(t) = \frac{\mathbf{q}_t^{init} \cdot \text{sample}(\mathbf{f}_t, \ \mathbf{p}_t)}{\mathbf{q}^{init} \cdot \text{sample}(\mathbf{f}_t, \ \mathbf{p}_t)} \tag{10}$$

Here, $\mathbf{p}_t$ represents the location of the ground-truth correspondence point, and $\mathbf{f}_t$ is the feature map of the target frame. On the DAVIS dataset, we calculated $s_{sr}$ for visible points, achieving a score of 1.24, indicating that spatial memory introduces a 24% increase in similarity compared to the initial feature. In Fig. 14, we visualize the similarity scores for different tracks over time for two videos from the DAVIS dataset. The plot highlights that the similarity increases more significantly toward the end of the video, where appearance changes are more severe. Moreover, the score is consistently greater than 1, showing that $\mathbf{q}_t^{init}$ always provides better initialization than $\mathbf{q}^{init}$ in these videos.

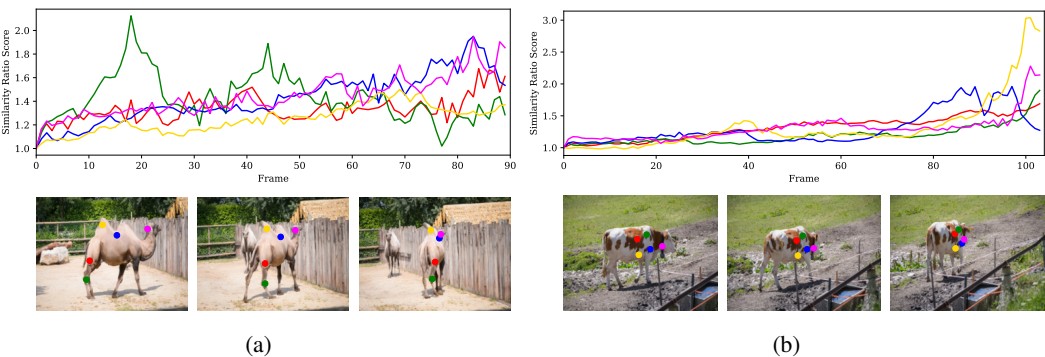

(a)  (b)

Figure 14: **Similarity Ratio Score.** The similarity ratio score $s_{sr} > 1$ over frames for different tracks, demonstrates increased similarity with ground-truth location on the target frame when utilizing spatial memory.

## F    FAILURE ANALYSIS

We identify three common failure cases: (i) tracking points on thin surfaces or edges, (ii) fast motion or scene cuts, and (iii) localization on uniform areas. We visualize examples of these failure cases,

where the average error is higher than 8 pixels for visible frames, in Fig. 15. In the visualizations, our predictions are represented as dots, while ground-truth correspondences are marked with diamonds. The line connecting the ground-truth and prediction indicates the error.

**Thin Surfaces:** Points of interest on thin surfaces and edges may not be well-represented in feature-level resolution due to the lack of pixel-level granularity. This limitation causes the model to track incorrect points by missing the actual point of interest. For instance, in the upper row of Fig. 15a, which shows an example from the DAVIS dataset, the model fails to track a rope and instead tracks the background, as the precision is insufficient to accurately represent the thin structure. Similarly, in the bottom row, from the Kinetics dataset, points of interest on a thin surface (*e.g.* a stick) are mislocalized, with the model tracking the background instead of the object.

**Fast Motion:** When the scene content changes rapidly, either due to fast camera motion or scene cuts (commonly seen in the Kinetics dataset), our model encounters difficulties in localizing previously tracked points. For instance, in the upper row of Fig. 15b (examples from Kinetics), the model fails to continue tracking after a large scale change, where the original view in the middle frame becomes significantly smaller in later frames. In the second row, the model struggles to resume tracking points after a considerable number of frames where the scene has completely changed due to a scene cut.

**Uniform Areas:** Localization on uniform or highly similar areas is more challenging, likely because most objects in the training dataset exhibit descriptive textures (Doersch et al., 2024). While our model can approximately localize points, it struggles with precision in these scenarios. This limitation is illustrated inFig. 15c, using an example from the RoboTAP dataset.

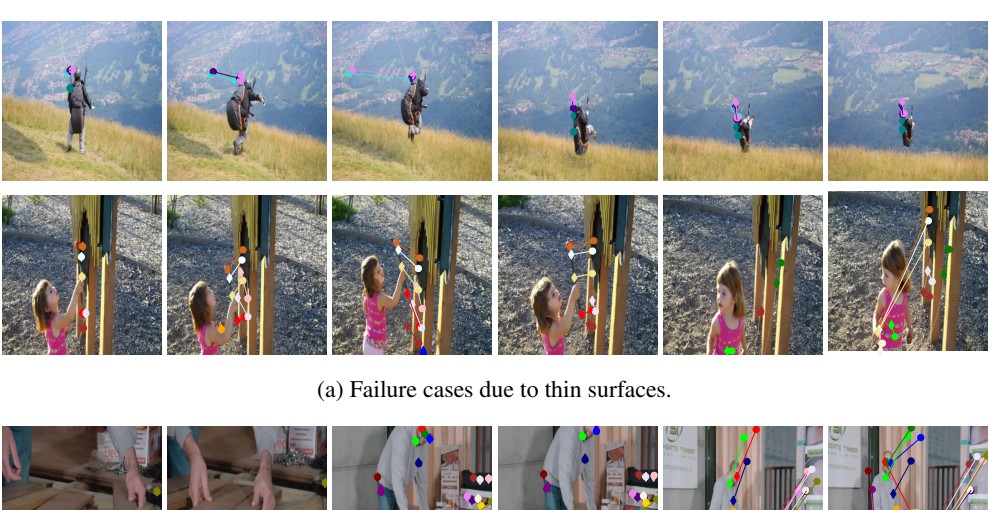

(a) Failure cases due to thin surfaces.

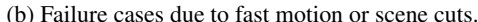

(b) Failure cases due to fast motion or scene cuts.

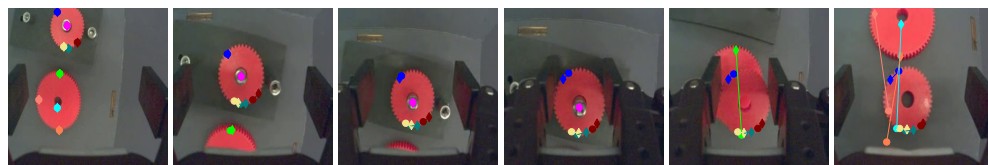

(c) Failure cases due to mis-localization on uniform areas.

Figure 15: **Common Failure Cases.** We identify three common failure cases: tracking points on thin surfaces (a), fast motion or scene cuts (b), and localization on uniform areas (c). We visualize predictions with average error higher than 8 pixels, where predictions are shown as dots and ground-truth correspondences are marked with diamonds. Different tracks are depicted in distinct colors.

