# OpenReview forum: "Track-On: Transformer-based Online Point Tracking with Memory"
_ICLR.cc/2025/Conference — ICLR 2025 Poster_

### Official Review · Reviewer_sUJJ · 2024-10-19

**Soundness:** 3
**Presentation:** 3
**Contribution:** 3
**Rating:** 6
**Confidence:** 5

**Summary:**

The paper proposes a simple yet effective framework, named Track-On, for Tracking Any Point (TAP) task.

Track-On employ a coarse-to-fine mechanism, where they first find the most match patch feature, and then predict an offset relative to the center of the most matched patch to obtain a more fine-grained prediction.

To tackle the issue of feature drifting in long-term situation, Track-On further proposes the Spatial Memory as well as Context Memory. Spatial Memory utilize deformable attention to flexibly select the most-relevant features in the last frame, to provide a more reliable and latest feature as a reference to help the prediction of current frame. While Context Memory maintains a FIFO queue to store the latest K frames' decoder outputs, to provide a larger context.

Extensive experiments are conducted. The performance on TAP-Vid benchmark shows their superiority, especially on Kinetics and RGB-Stacking subsets. The thorough ablation studies and supplementary materials make the paper looks solid.

**Strengths:**

1. The paper proposes a simple yet effective transformer-based framework for tracking any point task, named Track-On. Track-On employs the coarse-to-fine strategy. Track-On utilize the DINOv2's feature to help find the most matching feature patch to provide a coarse prediction. After that, Track-On predict a local offset to obtain a more fine-grained prediction. The coarse-to-fine strategy looks reasonable.

2. The paper proposes the Spatial Memory and Context Memory to handle the long-term tracking situation. Spatial and Context Memory provides complementary information for each other, improving their performance significantly.

3. Extensive experiments and ablation studies are conducted, showing their superiority and the effectiveness of each module.

**Weaknesses:**

I think there is no fatal weaknesses, but there are some concerns:

1. There are some designs that are similar with previous works. For example, in TAPIR, they also use coarse-to-fine strategy, in TAPTR, they also utilize the initial feature to alleviate error propagating. It is better to cite these paper during your description.

2. Some points are over-claimed. Actually, the Spatial/Context Memory can not "address" tracking drifting. There are more metric or experimental evidence to support the claims in L78-L87.

3. Since the positional embedding are learned during training process, how to guarantee the continuity of the embeddings? If the continuity can not be guaranteed, how to ensure the "inference-time memory extension"? What if just ignore the positional embedding?

**Questions:**

1. I wonder how the Fig. 4 is obtained. By sampling on each frame based on the gt position and then calculate the similarity?

2. It seems a typo in L289, what is the r^q?

**Details Of Ethics Concerns:**

There are no ethics concerns.

---

> ### Author Response · Authors · 2024-11-19
>
> We extend our sincere appreciation to the reviewer for their valuable insights and constructive feedback. We have responded to their questions and concerns, and we hope our explanations thoughtfully encompass the raised points.
>
> ## W1. Missing Citations
> We appreciate the reviewer highlighting the similarities with previous works. We have added citations in the paper (highlighted in blue), specifically in sections where we discuss the coarse-to-fine strategy (line 144) and the feature drift (line 215).
>
> ## W2. Analysis of Spatial Memory
> We have adjusted our claim in line 78 to clarify that Spatial Memory helps mitigate tracking drift rather than completely “addressing” it. Additionally, we conducted a thorough analysis of the effect of spatial memory on drift, detailed in **Appendix D** (highlighted in blue). We designed two metrics:
> * **Similarity Ratio Score**: This metric evaluates the similarity of the updated features to the ground truth features in the feature space. Our analysis showed that features updated using spatial memory have a higher similarity ratio score—improving by 20% over the initial query features—indicating better alignment with the target. Please refer to **Figure 12 in Appendix D** for a visual example illustrating how the similarity ratio score evolves over time across different tracks.
> * **Matching Score**: This evaluates tracking performance when using the updated query features directly for tracking. We found that in several cases, the initial query fails to track the target due to feature drift, whereas the updated features from the spatial memory successfully recover the track (**Figure 14a–c in the Appendix D**). Overall, the matching score improved by **32%** with the use of spatial memory and consistently enhanced tracking performance across all videos in the DAVIS dataset (see **Figure 13 in Appendix D**).
>
>     We also observed failure cases where spatial memory could not recover from drift, as shown in Figure 14d. This indicates that while spatial memory is effective in many scenarios, it may not fully address tracking drift in all cases.
>
> ## W3. Continuity of Temporal Positional Embeddings
> We present an analysis of the learned temporal positional embeddings in **Appendix E**. We utilize learnable temporal positional embeddings in our model to help it differentiate between different time steps, similar to the approach used in learnable spatial positional embeddings of ViTs:
> * **Guaranteeing Continuity:** To ensure continuity when extending the sequence length during inference, we apply linear interpolation to the learned temporal positional embeddings. This, similar to the resizing of positional embeddings in ViT, generates new embeddings that smoothly transition between learned positions, preserving temporal order.
>
>     _Example_: Given learned embeddings of size 3, $[p_0, p_1, p_2]$, we extend them to size 5 as $[p_0, \dfrac{p_0 + p_1}{2}, p1, \dfrac{p_1 + p_2}{2}, p_2]$. This linear interpolation maintains the relative positions and ensures continuity between embeddings.
>
> * **Importance of Positional Embedding:** We conducted an experiment to assess the impact of removing temporal positional embeddings, provided in Table 10 in Appendix E, and below for your reference. Without positional embeddings, the model’s performance drops by 1.8 points in AJ, 1.1 points in $\delta^{x}_{avg}$, and 2.1 points in OA. These results demonstrate that positional embeddings are crucial for capturing temporal dependencies.
>
> * **Visualization and Analysis:** We visualize the temporal positional embeddings by projecting them to 3D with PCA (Figure 15, **Appendix E**; in blue). We first show that ​the model learns temporal order, as seen from the color changing smoothly from the most recent to the oldest. Then, we extend the embeddings to longer time steps and show that the temporal order is preserved after interpolation.
>
> ## Q1. Explanation of Fig. 4 (Drift Plot)
> Yes, the reviewer’s understanding is correct. We sampled from the ground-truth position and calculated the similarity with the initial query feature. Formally, we plot the scaled dot product between the initial query $\mathbf{q}^{init}$ sampled from the query frame, and the features sampled from the ground-truth position $\mathbf{p}_t$ on the feature map ($\mathbf{f}_t$) at time t: $\mathbf{q}^{init} \cdot \text{sample}(\mathbf{f}_t, ~\mathbf{p}_t)$
>
> ## Q2. Typo
> Thank you for identifying this typo. We fixed it in the paper, $\gamma$ denotes the learnable, temporal positional embeddings. \
> &nbsp;
> ## Tables
> ### Table: Temporal Positional Embeddings
> | Temporal Embeddings | AJ   | $\delta^{x}_{avg}$ | OA  |
> |-----------|:------:|:------:|:-------:|
> | &cross; | 60.7  | 74.6    | 87.7  |
> | &check; | 62.5   | 75.7    | 89.8   |

---

> > ### Comment · Reviewer_sUJJ · 2024-11-25
> >
> > We thank the authors for the efforts made to address my concerns, which has consolidated my scoring decision. I hope these additional analysis can be included in the camera-ready version to make the paper more solid.

---

> > > ### Author Response · Authors · 2024-11-26
> > >
> > > We thank the reviewer for their positive assessment and constructive suggestions to strengthen the paper, which will be added to the final version of the paper, particularly the recommendation for additional analysis on spatial memory. We are happy to provide further details or clarifications to address any remaining concerns and, hopefully, improve the score.

---

### Official Review · Reviewer_D5B2 · 2024-10-27

**Soundness:** 3
**Presentation:** 2
**Contribution:** 2
**Rating:** 5
**Confidence:** 5

**Summary:**

In order to make point tracking methods more suitable for the real world, the author proposes a new frame by frame online tracking method. The author designed a spatial memory module and a content memory module to capture temporal information and maintain reliable point tracking over a long period of time. Finally, the author uses a coarse to fine manner to predict the final output.

**Strengths:**

1.The author designed two memory modules to mine spatio-temporal information and use coarse to fine manner for more accurate point prediction.

2.Experiments demonstrate the effectiveness of the method proposed by the authors and obtain the SOTA performance of the online point tracking method.

**Weaknesses:**

1.Experiments were inadequate. The designed analysis and the layer number ablation analysis of each modules are not enough, including several decoders, the number (4) of level of similar map used for patch classification, etc.

2.The figures in the paper are not clear enough. For example, in the left of Figure 5, the input qinit does not understand what it means, because q has no subscript and does not specify its meaning. In additional, the title of Figure 5 should go from left to right when introducing the content.

3.Does online point tracking just require that the model does not see the next frame during testing? I think the authors should add the speed of testing like SOT, as the authors said, to be more adaptable to the real world applications.

4.Although your online method is the best, it does not outperform the offline method in the DAVIS data set. What is the reason? Apart from the characteristics of the two type methods.

5. Although the author analyzed the causal processing method in the relevant video field, the analysis of the online method in point tracking is not sufficient, and it does not see the advantages of the proposed method in online methods.

**Questions:**

1.Although the authors demonstrated many visual maps to demonstrate the effectiveness of the method, a quantitative ablation analysis of the critical modules is lacking, for example, the layer number analysis of query decoder.

2.Many writing details should be noted by the author, including the expression of the picture content, the color of the formula and other details.

---

> ### Author Response · Authors · 2024-11-19
>
> We are grateful for the reviewer's feedback and perceptive remarks. We have provided responses and hope these will resolve any issues you've raised.
>
> ## W1./Q1. Ablation on Critical Modules
> Based on the reviewer’s suggestion, we conducted extensive ablation studies on critical modules of our model. Detailed results are presented in **Appendix C** (Fig. 11, Table 8), and below for your reference:
> * **Number of Layers in Query Decoder, Offset Head, and Memory Write Module**: We tested configurations with 1 to 4 layers. The best performance was achieved with 3 layers. For instance, increasing to 4 layers decreased occlusion accuracy (OA) by 0.8, indicating potential overfitting due to increased model complexity.
> * **Number of Scales**: We varied the number of scales from 1 to 4. The model with 4 scales outperformed others, confirming the effectiveness of multi-scale features in capturing spatial information.
>
> Due to time constraints, these models were trained for 50 epochs instead of the full 75, but the trends provide valuable insights. Please  refer to Appendix C for more detailed analysis.
>
> ## W2./Q2. Fixes in the Paper
> We have clarified the notation and revised the caption of Figure 5 to introduce the content from left to right (highlighted in blue). Additionally, we clarified the color of the equation 9 (line 310)
>
> ## W3. Speed of Online Tracking
> Run-time speed is critical to online point tracking applications. If we understand correctly, we already reported Track-On's inference speed in Fig. 7 of the main paper by varying the memory size. While large enough memory is critical to accuracy (Fig. 6), our method offers reasonable trade-offs between accuracy (AJ; Fig 6) and speed (FPS; Fig. 7) with real-time options (~20 FPS).
>
> ## W4. Comparison between Online and Offline
> While a performance gap is expected due to the inherent advantages of offline methods utilizing future information, our method nonetheless achieves competitive results—even surpassing offline methods on some datasets—while operating under stricter constraints.
>
> ## W5. Analysis of Other Online Methods
> While only a few existing methods, namely TAPIR, BootsTAPIR, and DynOMo, report results in the online setting, they are not specifically designed for online processing.
> * Combining existing offline methods, **TAPIR** performs iterative updates (PIPs), starting from a rough initial estimate (TAP-Net). The online version of TAPIR was first introduced in RoboTAP for robotics tasks that require online processing. To adapt TAPIR for online use, it is re-trained with a temporally causal mask, processing frames sequentially on a frame-by-frame basis.
>
> * **BootsTAPIR** is a scaled-up version of TAPIR, trained on 15M real images to enhance performance. However, it still follows the same underlying approach and is not inherently designed for online operation.
>
> * **DynOMo** introduces an expensive test-time optimization method, which takes approximately 45 seconds to process a single frame. This significant computational cost makes it impractical for real-time applications that require prompt responses.
>
> * We have also evaluated other offline methods, e.g. **CoTracker** and **SpatialTracker**, under the online setting, as explained in our paper (lines 360–366). However, these methods are not designed for online processing.
>
> * In contrast, our proposed method **Track-On is specifically designed for the online setting**. We introduce novel memory modules that effectively capture temporal information, enabling accurate point tracking without relying on future frames. We clarified these points in the related work section, lines 492–496, highlighted in blue.
>
> We hope these clarifications address your concerns. Thank you once again for your valuable feedback, which has been instrumental in improving our work. \
> &nbsp;
> ## Tables
> ### Table: Query Decoder Layer Number
> | #Layers | AJ   | $\delta^{x}_{avg}$ | OA  |
> |-----|:------:|:------:|:-------:|
> | 1 | 60.3   | 74.4    | 88.2   |
> | 2 | 60.9   | 75.2    | 88.8   |
> | 3 | 62.1   | 75.3    | 89.6   |
> | 4 | 61.6   | 75.2    | 88.8   |
> &nbsp;
> ### Table: Offset Head Layer Number
> | #Layers | AJ   | $\delta^{x}_{avg}$ | OA  |
> |-----|:------:|:------:|:-------:|
> | 1 | 61.8   | 75.6    | 88.9   |
> | 2 | 61.1   | 74.9    | 88.9   |
> | 3 | 62.1   | 75.3    | 89.6   |
> | 4 | 61.0   | 75.0    | 88.0   |
> &nbsp;
> ### Table: Query Memory Writer Layer Number
> | #Layers | AJ   | $\delta^{x}_{avg}$ | OA  |
> |-----|:------:|:------:|:-------:|
> | 1 | 61.5  | 75.6    | 89.5   |
> | 2 | 61.3   | 74.9    | 89.4   |
> | 3 | 62.1   | 75.3    | 89.6   |
> | 4 | 62.3   | 75.2    | 89.1   |
> &nbsp;
> ### Table: Number of Scales
> | #Scales | AJ   | $\delta^{x}_{avg}$ | OA  |
> |-----|:------:|:------:|:-------:|
> | 1 | 61.4  | 74.9    | 88.6  |
> | 2 | 61.3   | 74.7    | 88.8   |
> | 3 | 61.0   | 74.7    | 88.7   |
> | 4 | 62.1   | 75.3    | 89.6   |

---

> ### Author Response · Authors · 2024-11-26
>
> We thank the reviewer for their constructive feedback, including the suggestion for additional ablations to provide deeper insights and a clearer comparison with other online models. We kindly ask if our rebuttal has sufficiently addressed your concerns and if the score might be reconsidered.

---

### Official Review · Reviewer_BkZB · 2024-10-30

**Soundness:** 3
**Presentation:** 3
**Contribution:** 3
**Rating:** 8
**Confidence:** 4

**Summary:**

This paper proposes a long-term point tracking framework focused on online, frame-by-frame tracking. The framework leverages spatial and context memory modules to reliably track points over extended time horizons. It achieves state-of-the-art results for long-term point tracking in the online setting. Additionally, the framework is capable of real-time tracking.

**Strengths:**

- The paper is well-written and easy to follow, with extensive experiments that clearly demonstrate the performance improvements contributed by each module.
- The authors propose an effective approach using patch classification, rather than regression, to predict each point location; while patch classification was mentioned in the PIPs paper, it was previously used only to accelerate convergence by supervising score maps.
- The authors propose 2 memory modules that effectively improve the tracking performance.

**Weaknesses:**

- line 289: I think you meant $\gamma^s$ not $\gamma^q$ which does not appears in eq.8.
- Table 1: the authors should show the backbone used in each approach for easier comparison. If possible, the methods should be compared on the same backbone. However, this is alleviated by table 3 where the authors have show the effectiveness of the memory modules.

**Questions:**

It would be interesting if the authors can show that this method can be used on much longer videos from recent datasets such as PointOdyssey or Tapvid3D.

---

> ### Author Response · Authors · 2024-11-19
>
> We sincerely thank the reviewer for their thoughtful feedback and have addressed each point below.
>
> ## W1. Typo
> We appreciate the reviewer pointing out the typo in line 289 (now, 291). We have corrected the mistake in the revised version, highlighted in blue.
>
> ## W2. Backbone Comparison
> Following the reviewer’s suggestion, we conducted additional experiments where we replaced the backbone of our model (ViT-Adapter) with the Residual CNN used by CoTracker. The results of these experiments, along with the number of learnable parameters and backbones for each model, are presented in **Appendix C** (Table 9), and below for your reference. Our findings are summarized as follows:
> * **Performance with ViT-Adapter Backbone:**  Our default model, utilizing the ViT-Adapter backbone, achieves the highest performance among the compared methods. Notably, it does so with significantly fewer parameters, approximately half that of CoTracker (23M vs. 45M) and one-third of BootsTAPIR (78M).
>
> * **Performance with Residual CNN Backbone:** When we switched to CoTracker’s Residual CNN backbone, our model experienced a slight performance decrease (a reduction of 1.3 in AJ). However, even with this backbone, our model still outperforms other methods with more parameters, including CoTracker itself (our model: 61.6 vs. CoTracker: 55.9 in AJ).
>
> These results demonstrate that our method’s superior performance is not solely attributable to the choice of backbone but is also due to our novel approach and effective use of the memory module. The comparison reinforces the robustness and generalizability of our method across different backbone architectures.
>
> ## Q1. Longer Videos
> We appreciate the reviewer’s suggestion to evaluate our method on longer video sequences from recent datasets. To address this, we have started incorporating the PointOdyssey dataset into our evaluation. We have reached out to the authors of CoTracker for guidance on data preprocessing and evaluation, and are currently awaiting their response. We will include these results in the final version of the paper or provide them during the rebuttal phase once we receive the necessary files.\
> &nbsp;
> ## Tables
> ### Table: Backbone and Number of Learnable Parameters
> | Model              | Backbone            | #L.P. | AJ   |
> |--------------------|---------------------|:-----:|:------:|
> | TAPIR             | ResNet-18          | 29M   | 56.7   |
> | CoTracker (All)   | Residual CNN       | 45M   | 55.9   |
> | BootsTAPIR        | ResNet-18 + 4 Conv | 78M   | 59.7   |
> | *Ours*            | Residual CNN       | 41M   | 61.6   |
> | *Ours*                | ViT-Adapter        | 23M   | 62.9   |

---

### Official Review · Reviewer_miCL · 2024-11-03

**Soundness:** 3
**Presentation:** 3
**Contribution:** 3
**Rating:** 5
**Confidence:** 4

**Summary:**

The paper introduces Track-On, a transformer-based online long-term point tracking model designed for real-time point tracking in video streams. Track-On leverages two memory modules—spatial memory and context memory—to capture temporal information and maintain reliable point tracking over extended video sequences. Unlike methods relying on iterative updates and full temporal modeling, Track-On employs patch classification and refinement for identifying correspondences and tracking points with high accuracy. Extensive experiments demonstrate that Track-On sets a new state-of-the-art for online models and delivers superior or competitive results compared to offline approaches on the TAP-Vid benchmark.

**Strengths:**

The dual memory module design effectively handles long video sequences in an online setting while addressing feature drift.

Track-On demonstrates fast inference speeds while maintaining high accuracy, a significant advantage in real-time tracking systems.

The model establishes a new benchmark for online models on the TAP-Vid benchmark and shows competitive performance against offline methods.

**Weaknesses:**

The paper mentions potential precision loss when tracking thin surfaces or instances with similar appearances, indicating limitations in handling certain visual effects.

The model may struggle to establish correct correspondences in complex scenes with high visual similarity, affecting tracking accuracy.

While the model is memory-efficient, there is a trade-off between memory module size and inference speed, which may require balancing in practical applications.

**Questions:**

There are too few datasets for comparison, I hope to supplement the experiments.

Are there common patterns or specific scene characteristics in the cases where the model fails?

There are other transformer-based tracking methods mentioned, such as SpatialTracker and CoTracker. How does Track-On differentiate itself from these methods?

---

> ### Author Response · Authors · 2024-11-19
>
> We thank the reviewer for their insightful feedback and have addressed each point below. We hope that these clarifications address the reviewer’s concerns.
>
> ## W1. /Q2. Common Failure Cases
>  We have identified two common failure cases:
> * **Points on Thin Surfaces:** Tracking accuracy diminishes on thin surfaces due to limited spatial information. Increasing the feature resolution might alleviate this issue, although it would require additional memory overhead.
> * **Points in Uniform Areas:** Uniform regions lacking distinctive visual features pose a significant challenge for point tracking. The absence of descriptive cues makes it difficult for the model to establish accurate correspondences. Training on large-scale real-world datasets can mitigate these challenges, as demonstrated by BootsTAPIR’s improved performance resulting from training on 15M real images.
>
> We have included examples illustrating these failure cases across multiple datasets in **Appendix F** (highlighted in blue).
> ## Q1. Results on Additional Datasets
> Following the reviewer’s suggestion, we have evaluated our model on two additional datasets: RoboTAP and Dynamic Replica. The results are presented in **Appendix B** (Tables 4 and 5; highlighted in blue), and below for your reference. Our method outperforms all other models on these datasets, including offline methods, except for BootsTAPIR on RoboTAP in certain metrics (in AJ and $\delta^{x}_{avg}$). The superior performance of BootsTAPIR in those metrics is likely due to its extensive training on a non-public dataset of 15 million real images. Notably, our model surpasses all other models that were trained on publicly available dataset TPVid-Kubric. These additional experiments reinforce the robustness and generalizability of our approach across diverse scenarios on 6 datasets in total.
>
> ## W2. Trade-off Between Memory Size and Inference Speed
> As the reviewer correctly noted, there is a trade-off between memory size and inference speed. While this might seem like a weakness, it actually provides flexibility to choose models with varying capacities and speeds, including some real-time options (~20 FPS). Our analysis also shows that memory needs to function as a bottleneck because increasing memory size decreases performance beyond a certain point. One future direction could be to learn how to compress large temporal information before storing it in memory, to obtain faster inference even when storing more information in memory.
>
> ## Q3. Differences with Other Transformer-Based Trackers
> Our method fundamentally differs from other transformer-based tracking approaches like CoTracker and SpatialTracker, especially in how correspondences are established:
> * **Previous Methods**: CoTracker and SpatialTracker process all frames within a temporal window simultaneously. They use transformers to **iteratively update point features and correspondence estimations** by sharing spatial and temporal information across frames. After several iterations, they regress the point coordinates based on these updated features. This approach requires multiple passes and access to all frames in the window, making it less suitable for real-time applications.
> * **Our Approach**: We treat points of interest as **queries in a transformer decoder** and directly decode them to establish correspondences on a **frame-by-frame basis**. During training, we employ a classification loss, which enables our model to predict correspondences without the need for iterative updates. This allows our method to process each frame independently, without relying on future frames or multiple iterations.This key difference enables our method to **operate online**, processing frames sequentially as they arrive. This is necessary for real-time applications or scenarios where future frames are not available during inference. In contrast, methods that depend on iterative updates over multiple frames are not suitable for such situations because they require access to a window of frames. \
> &nbsp;
> ## Tables
>
> ### Table: Results on Dynamic Replica
> | Model               | Online | $\delta^{vis}_{avg}$ |
> |---------------------|:------:|:--------:|
> | TAP-Net | &check;     | 53.3     |
> | PIPs| &cross;      | 47.1     |
> | TAPIR| &cross;     | 66.1     |
> | CoTracker (Single) | &cross;     | 68.9     |
> | BootsTAPIR| &cross;     | 69.0     |
> | TAPTR              | &cross;     | _69.5_   |
> | *Ours*             | &check;    | **72.7** |
> &nbsp;
> ### Table: Results on RoboTAP
> | Model               | Online     | AJ   | $\delta^{x}_{avg}$ | OA  |
> |---------------------|:----------:|:------:|:-------:|:------:|
> | TAP-Net            | &check; | 45.1| 62.1| 82.9|
> | TAPIR              | &cross; | 59.6| 73.4| _87.0_ |
> | CoTracker (Single) | &cross; | 52.0| 65.5    | 78.8 |
> | BootsTAPIR         | &cross;| **64.9** | **80.1** | 86.3 |
> | TAPTR              | &cross;| 60.1   | 75.3 | 86.9   |
> | *Ours*             | &check;| _62.2_ | _75.8_ | **88.8** |

---

> ### Author Response · Authors · 2024-11-26
>
> We thank the reviewer for their constructive feedback, including the suggestion to enhance the paper with two additional datasets and a failure analysis. We kindly ask if our rebuttal has addressed your concerns and whether the score could be adjusted accordingly.

---

> ### Author Response · Authors · 2024-11-29
>
> Dear reviewer miCL,
>
> We have conducted an additional evaluation of our model using the PointOdyssey dataset, which includes longer videos (~2000 frames). For detailed insights, please refer to our response to Reviewer BkZB. We hope that this extensive evaluation across **seven datasets** with diverse characteristics sufficiently addresses your concerns.

---

### Author Response · Authors · 2024-11-19

Dear reviewers,

We are grateful for your evaluation of our submission and your detailed feedback. We uploaded an updated version with additional results and fixes according to your comments.

In addition to specific, detailed responses to each reviewer, we provide a list of changes here as a summary. Please refer to the specified parts in the paper and the Appendix for the details:

* **Fixes to the main paper (BkZB, D5B2, suJJ)**: Revisions to captions, additional explanations, and completing missing references (in blue).
* **Evaluation of two additional datasets (miCL)**: Results on RoboTAP and Dynamic Replica (Appendix B).
* **Ablating hyper-parameters (D5B2)**: Ablating the number of scales, the number of layers in the query decoder, offset head, and spatial memory writer (Appendix C).
* **Changing the backbone (BkZB)**: Replacing the backbone with the feature encoder from CoTracker (Appendix C).
* **Analysis of spatial memory (suJJ)**: Introducing two metrics to evaluate the effect of spatial memory on feature drift with visualizations (Appendix D).
* **Visualization of temporal embedding (suJJ)**:  Visualization of learned temporal embeddings to provide insight into what is captured by these embeddings (Appendix E).
* **Removing temporal embedding (suJJ)**: Evaluating the impact of removing temporal positional embeddings during training (Appendix E).
* **Visualizing common failure cases (miCL)**: Grouping common failure patterns and providing visual examples (Appendix F).

We have provided detailed clarifications, and hope these can resolve the reviewers’ concerns, and thus help raise their scores accordingly.

Thank you for your consideration.

---

### Author Response · Authors · 2024-11-21
**Kind Reminder**

Dear Reviewers,

We would appreciate it if you could share with us whether our rebuttal has addressed your concerns. We would also be happy to answer if you have any further questions.

Thank you.

---

> ### Author Response · Authors · 2024-11-24
> **Kind Reminder**
>
> Dear reviewers,
>
> As the discussion period nears its conclusion, we kindly ask for a response to our rebuttal. We are happy to provide further clarifications.
>
> Thank you.

---

### Meta-Review · Area_Chair_t8Zw · 2024-12-20

**Metareview:**

Summary:
This paper proposes a simple and efficient transformer-based framework for tracking any point task. The proposed method leverages spatial and context memory modules to reliably track points over extended time horizons. The experimental results on DAVIS and Kinetics datasets are given.

The main strengths include 1) strong performance, and 2) well-motivated and easy to follow.
The main weaknesses include 1) lack of detailed ablation studies, 2) missing speed evaluation, and 3) lack of evaluations on more challenging cases and comparison against more recent methods.

After the discussion phase, the main issues are lack of comparison with recent works and explanations of the experiments.
During the rebuttal phase, the authors provided more experiments (3 additional datasets, more failure cases, and more ablation studies) and clarifications, which address the main issues of lack of extensive evaluation and analysis (recognized by two of the reviewers, the other two did not respond).
Considering the innovative idea and good performance, the AC decided to go with sUJJ and BkZB’s recommendation to accept the paper, but the authors should consider the reviewers' suggestions to include necessary experimental comparisons/analyses.

**Additional Comments On Reviewer Discussion:**

The main concerns raised by the reviewers are lack of extensive evaluation and analysis, including more detailed ablation study, speed evaluation, more challenging case study, and more comparisons against recent methods.
During the rebuttal, the authors provided more experiments (3 additional datasets, more failure cases, and more ablation studies) and clarifications.

Two reviewers participated in the discussion and agreed that the authors' feedback addressed the issues they raised. However, the other two reviewers did not respond.
The final ratings come to 8655.

---

### Decision · Program_Chairs · 2025-01-22

Accept (Poster)